# A New Generation of IMiDs as Treatments for Neuroinflammatory and Neurodegenerative Disorders

**DOI:** 10.3390/biom13050747

**Published:** 2023-04-26

**Authors:** Katherine O. Kopp, Margaret E. Greer, Elliot J. Glotfelty, Shih-Chang Hsueh, David Tweedie, Dong Seok Kim, Marcella Reale, Neil Vargesson, Nigel H. Greig

**Affiliations:** 1Drug Design & Development Section, Translational Gerontology Branch, Intramural Research Program National Institute on Aging, Biomedical Research Center, 251 Bayview Blvd., NIH, Baltimore, MD 21224, USA; 2Faculty of Medicine, Georgetown University School of Medicine, Washington, DC 20007, USA; 3Department of Neuroscience, Karolinska Institutet, 17177 Stockholm, Sweden; 4Aevisbio Inc., Gaithersburg, MD 20878, USA; 5Aevis Bio Inc., Daejeon 34141, Republic of Korea; 6Department of Innovative Technologies in Medicine and Dentistry, G. d’Annunzio University of Chieti and Pescara, 66100 Chieti, Italy; 7School of Medicine, Medical Sciences and Nutrition, Institute of Medical Sciences, University of Aberdeen, Aberdeen AB25 2ZD, UK

**Keywords:** immunomodulatory imide drugs (IMiDs), thalidomide, pomalidomide, neuroinflammation, cereblon, traumatic brain injury, Alzheimer’s disease, Parkinson’s disease, neurodegeneration, erythema nodosum leprosum

## Abstract

The immunomodulatory imide drug (IMiD) class, which includes the founding drug member thalidomide and later generation drugs, lenalidomide and pomalidomide, has dramatically improved the clinical treatment of specific cancers, such as multiple myeloma, and it combines potent anticancer and anti-inflammatory actions. These actions, in large part, are mediated by IMiD binding to the human protein cereblon that forms a critical component of the E3 ubiquitin ligase complex. This complex ubiquitinates and thereby regulates the levels of multiple endogenous proteins. However, IMiD-cereblon binding modifies cereblon’s normal targeted protein degradation towards a new set of neosubstrates that underlies the favorable pharmacological action of classical IMiDs, but also their adverse actions—in particular, their teratogenicity. The ability of classical IMiDs to reduce the synthesis of key proinflammatory cytokines, especially TNF-α levels, makes them potentially valuable to reposition as drugs to mitigate inflammatory-associated conditions and, particularly, neurological disorders driven by an excessive neuroinflammatory element, as occurs in traumatic brain injury, Alzheimer’s and Parkinson’s diseases, and ischemic stroke. The teratogenic and anticancer actions of classical IMiDs are substantial liabilities for effective drugs in these disorders and can theoretically be dialed out of the drug class. We review a select series of novel IMiDs designed to avoid binding with human cereblon and/or evade degradation of downstream neosubstrates considered to underpin the adverse actions of thalidomide-like drugs. These novel non-classical IMiDs hold potential as new medications for erythema nodosum leprosum (ENL), a painful inflammatory skin condition associated with Hansen’s disease for which thalidomide remains widely used, and, in particular, as a new treatment strategy for neurodegenerative disorders in which neuroinflammation is a key component.

## 1. Introduction

A new generation of immunomodulatory imide drugs (IMiDs) have emerged as treatments for neurodegeneration and pharmacological tools for understanding the role of neuroinflammation in the progression of neurodegenerative disease. One of the primary mechanisms of action of IMiDs is their ability to block the translation of tumor necrosis factor-alpha (TNF-α) mRNA into protein, thus blunting the production of one of the most prominent pro-inflammatory cytokines. Current US Food and Drug Administration (FDA)-approved IMiDs, including thalidomide, lenalidomide, and pomalidomide, are used as anticancer drugs, particularly in the treatment of multiple myeloma, as well as therapy for Hansen’s disease (leprosy and related symptoms). Despite their benefits, these drugs have been historically controversial due to their use between 1958–1962 by pregnant women, which resulted in teratogenic effects in their resulting children. These complications have stunted potential repurposing of IMiDs, including as a therapeutic for neurodegenerative disorders. Recent generations of IMiD-like drugs are chemically altered to provide the anti-neuroinflammatory benefits potentially without the associated adverse effects, including teratogenicity and peripheral neuropathy. These novel compounds may, therefore, afford a safer potential treatment option for neurodegeneration. We will highlight studies of three of these IMiD-like compounds selected from agents synthesized and pharmacologically evaluated by our research team to target neuroinflammation. The scarcity of effective treatments for neurodegenerative disease progression creates a critical current need to identify new drugs—in response, we propose combatting neurodegeneration through targeting neuroinflammation, utilizing a new class of IMiD compounds with a greater safety margin.

## 2. Neuroinflammation and Neurodegeneration: A Feed-Forward Cycle

Chronic neuroinflammation is common to neurodegenerative diseases, such as Alzheimer’s disease (AD) [1,2,3] and Parkinson’s disease (PD) [4,5,6,7], as well as to acute neural disorders, such as traumatic brain injury (TBI) [8] and ischemic stroke [9]. Albeit a key component of the innate immune system and endogenous reparative mechanisms [10,11,12,13], chronic or excessive neuroinflammation can lead to dysregulation and promote the progression of neurodegenerative disease [14]. As the disease progresses, consequences of neuroinflammation, such as synaptic loss, are associated with the progressive loss of established central nervous system (CNS) functions, including mobility, coordination, memory, and learning [15]. In turn, all neurodegenerative disorders are accompanied by several factors that trigger inflammation, including, but not limited to, aggregation or accumulation of certain pathogenic or abnormally modified proteins, genetic abnormalities, and the release of damage-associated molecular patterns (DAMPs), cell debris, and reactive oxygen species (ROS) from damaged cells (Figure 1) [16,17,18]. Pro-inflammatory intercellular messengers, such as the canonical cytokine TNF-α, interleukin (IL)-1β, and IL-6, mediate interactions between numerous cell types (including neurons, microglia, and astrocytes) that drive neurodegenerative disease progression [17,19,20]. Aberrant or prolonged neuroinflammation can create a positive feedback loop of inflammation, compounding damage to the CNS and further exacerbating neurodegenerative disease [21].

In the presence of deleterious stimuli, such as CNS injury or dysregulation, the brain’s immune cells, microglia, are activated into a spectrum of phenotypes that includes a highly pro-inflammatory state (Figure 2A) [19] and a homeostatic or quiescent state, where microglia serve as the immune cells of the brain, monitoring tissue for injury or abnormal signaling [22]. Shifts from a quiescent surveying state to activated states, commonly referred to as pro-inflammatory (M1) and alternatively activated/anti-inflammatory (M2) states, occur rapidly during acute brain injury, and these states are prolonged in chronic neurodegenerative conditions [23]. Previously activated microglia may be primed for future deleterious events and retain a proinflammatory phenotype. Primed microglia produce excess cytokines upon reactivation (Figure 2B) [24,25]. It is important to note that, although the M1 and M2 classifications are useful for descriptive purposes, microglial function should not be considered dichotomous, as a spectrum of microglial activation states has been revealed in single cell sequencing studies, and our understanding of this spectrum continues to expand [24]. 

Among the most widely influential pro-inflammatory cytokines is TNF-α, which is released primarily by microglia in the CNS and by monocytes in the periphery in response to deleterious stimuli (Figure 3) [26,27,28,29]. In mice and in humans, most CNS cell types (including neurons, astrocytes, and microglia) express TNF receptors (TNFR)-1 and 2, with the most prominent expression in astrocytes and microglia [30,31]. At physiological levels, TNF-α regulates numerous healthy neuronal functions, including excitatory neurotransmission, trafficking of α-amino-3-hydroxy-5-methyl-4-isoxazolepropionic acid (AMPA) receptors, homeostatic synaptic scaling, long term potentiation (LTP), and neurogenesis [32,33]. After injury, TNF-α is involved in neuronal recovery and initiates repair mechanisms [34]. However, chronic and/or excessive elevation of TNF-α levels fosters a neuroinflammatory environment that becomes detrimental to neuronal health, largely through stimulation of the nuclear factor kappa-light-chain-enhancer of activated B cells (NF-κB) pathway [20,35]. TNFR1 stimulation intracellularly activates NF-κB, a family of inducible transcription factors, which leads to increased expression of inflammation-related genes, including those for chemokines and cytokines, such as TNF-α [36]. This creates a feed-forward loop, as these inflammatory factors act in autocrine and paracrine fashions, and their activity can persist chronically without resolution [37]. Often, pharmacological interventions are necessary to break the cycle, reduce generation of TNF-α and other pro-inflammatory cytokines, and resolve the inflammatory response. Thus, TNF-α is a promising pharmacological target of neuroinflammatory drugs and potential treatments for neurodegeneration [38].

Microglia are not the only drivers of the neuroinflammatory feed-forward loop; astrocytes are implicated, as well. Activated microglia release a variety of inflammatory chemokines and cytokines, including TNF-α, IL-1α, and complement component 1 subcomponent q (C1q) that, when combined, directly act on astrocytes and promote the switch to their reactive and neurotoxic phenotypes, thereby further feeding neuroinflammation [39]. To illustrate, knockout of microglial-derived IL-1α, TNF-α, and C1q—pro-inflammatory astrocytic activators—in a genetic mouse model of amyotrophic lateral sclerosis (ALS) increased animal life span and improved peripheral nerve fiber formation when compared to control [40]. In another study with a mouse model of PD, administration of an anti-inflammatory glucagon-like peptide-1 (GLP-1) receptor agonist was found to dampen IL-1α, TNF-α, and C1q levels, as well as the formation of neurotoxic reactive astrocytes [41]. Upon their initial discovery in pathological contexts, reactive astrocytes were categorized as A1 (damaging/neurotoxic) and A2 (reparative) phenotypes, but these states exist on a spectrum similar to that of microglia and have a variety of context-dependent phenotypes, favoring a more nuanced nomenclature [42]. For descriptive purposes, we will utilize the A1 and A2 nomenclature herein. A1 astrocytes destroy synapses and are neurotoxic, exacerbating the neuroinflammatory environment and preventing neuronal repair; on the other hand, A2 astrocytes upregulate neurotrophic factors and have demonstrated neuroprotective functions [39,41,43,44,45]. The thorough utilization of immunochemical and genetic analysis techniques has revealed that A1 astrocytes are elevated in a variety of neurodegenerative diseases, including AD, PD, multiple sclerosis (MS), ALS, and Huntington’s disease (HD) [39]. The pro-inflammatory molecules released by A1 astrocytes (as well as M1 microglia) interact with neurons and promote neurodegeneration [44,46]. Thus, the interplay between microglial- and astrocytic-derived inflammatory factors and other CNS cell types is implicated in neurodegenerative disease progression. 

## 3. Anti-Inflammatory Therapies to Treat Neurodegenerative Disease

With reducing neuroinflammation as a promising drug target to mitigate neurodegeneration becoming recognized, numerous research groups have investigated ‘traditional’ anti-inflammatory drugs as potential treatments for neurodegenerative disease. Epidemiological studies and clinical trials yield mixed results on the use of non-steroidal anti-inflammatory drugs (NSAIDs) in slowing AD progression, with epidemiological studies tending to find benefits of NSAID use and randomized clinical trials tending to find no effect [47,48,49,50,51,52]. In one of several meta-analyses of cohort studies of AD patients taking NSAIDs, NSAID exposure was significantly associated with reduced risk for AD [53]. On the other hand, as one of numerous examples, in a recent randomized clinical trial, an anti-inflammatory antibiotic failed to exhibit significant effects in delaying AD progression [54]. As existing traditional anti-inflammatory treatments that have been tested thus far in AD patients have proven to be largely inadequate in improving cognitive and behavioral outcomes, studies of IMiDs were motivated as a novel anti-inflammatory drug strategy to mitigate neurodegeneration.

Studies of acute neural injuries, such as TBI or stroke, shed additional light on neuroinflammatory mechanisms and preventions, as these conditions are accompanied by severe neuroinflammation. Additionally, preclinical animal models of TBI are useful tools to evaluate neuroinflammatory mechanisms that are also common in human TBI [8]. Despite this, there remains a lack of treatments for TBI, similar to the dearth of drugs for treating chronic neurodegenerative diseases [55]. This failure could perhaps be explained by the heterogeneous nature of these injuries. Nevertheless, microglial activation and consequent mass cytokine release are clearly evident in both TBIs and chronic neurodegenerative diseases [54,56], and microglial phenotypic screening could be a better way to find useful compounds [57,58] that ameliorate crucial TBI parameters, such as behavioral impairments, neuronal death, and microglial phenotypic alteration. Cytokine-suppressive anti-inflammatory drugs (CSAIDs) may have greater treatment potential than NSAIDs for neuroinflammation and neurodegeneration.

In this context, traditional immunosuppressants that target TNF-α, epitomized by the biological medications infliximab, adalimumab, etanercept, golimumab, and certolizumab, have had profoundly beneficial effects in a variety of autoimmune diseases. However, these monoclonal antibody (mAb) drugs are proteins that lack oral deliverability and have limited blood–brain barrier (BBB) permeability for use in neurological disorders. In order to enter the cerebrospinal fluid (CSF) and brain, such TNF-α binding and clearing mAb drugs require to be administered via perispinal injection with Trendelenburg positioning, where they have demonstrated favorable results [26,59,60,61]. In contrast, IMiDs are orally deliverable, have reasonable bioavailability, and can penetrate the BBB, potentially making them more accessible and effective treatments for CNS diseases [21].

## 4. IMiDs

### 4.1. IMiD Signaling

IMiDs produce diverse pharmacological effects of potential clinical value for specific diseases, and they signal through several pathways (Figure 4). One key pathway that has defined earlier generation IMiDs is initiated by binding to cereblon. Cereblon is the substrate receptor of the ubiquitin-E3 ligase complex, which is comprised of DNA damage-binding protein (DDB1), cullin-4 (CUL4), and regulator of cullins-1 (ROC1). This complex serves to ubiquinate proteins that require to be targeted for degradation [62,63]. Under normal conditions, cereblon binds and targets only specific proteins for degradation, and it does not recognize others. However, when an IMiD attaches to cereblon through its glutarimide moiety, it allosterically modulates cereblon’s IMiD binding domain, allowing binding and degradation of a different set of proteins [64]. This action of older generation IMiDs is associated with the compounds’ observed teratogenicity and other adverse effects that are considered mediated via ubiquitination/degradation of key transcription factor proteins, but it additionally appears to underpin the anticancer, anti-angiogenic, and immunoregulatory effects of these same IMiDs [62,65,66,67]. 

The anti-inflammatory effects that classical IMiDs exert by binding to cereblon may occur through ubiquitin-dependent and ubiquitin-independent pathways [62,68,69], as well as by cereblon-independent pathways. In this regard, it has been known since the 1990s that one of the main effects of thalidomide is to decrease the TNF-α mRNA half-life [70] and, hence, the amount of protein that is translated from it, which explains some of thalidomide’s anti-inflammatory effects. Using cereblon gene (CRBN) knockdown, it has additionally been demonstrated that the inhibitory effect of IMiDs on TNF-α generation persisted in the silencing of CRBN [71,72], highlighting a cereblon-independent mechanism of anti-inflammation. Likewise, cereblon-independent pathways may potentially exist and underlie the other pharmacological actions of thalidomide-like drugs [63,73,74].

In the ubiquitin-dependent pathway, when a classical IMiD binds to cereblon (Figure 4A), it recruits the Cys_2_-His_2_ (C2H2) zinc finger domain and allows the ubiquitin-E3 ligase complex to ubiquitinate and degrade the neosubstrates Ikaros (Ikaros zinc finger (IKZF) 1)) and Aiolos (IKZF3) that are both members of the Ikaros family of zinc finger transcription factors for differentiation of B and T cells [75,76]. Degradation of Ikaros and Aiolos leads to suppression of interferon regulatory factor 4 (IRF4), a transcription factor critical for myeloma survival, as well as elevation of IL-2, an enhancer of natural killer cells, which supports attack on cancer cells and, thereby, inhibits tumorigenesis [62,68,71,77,78]. Studies in myeloma cell lines have revealed that knockout of Ikaros and Aiolos produces antitumor effects, highlighting a mechanism for the anticancer effects of IMiDs [67]. Furthermore, in multiple myeloma patients treated with IMiDs, Ikaros expression was reduced in macrophages, which promoted tumoricidal activity [79]. 

With regard to the ubiquitin-independent pathway, IMiDs compete with cereblon to bind to the CD147-MCT1 complex (Figure 4B) [69]. The CD147-MCT1 complex is involved in numerous functions that enhance tumor survival, including proliferation, invasion, angiogenesis, cell survival, and tumor aggressiveness [69]. When IMiDs compete with cereblon for binding to the CD147-MCT1 complex, destabilization of the complex occurs, which inhibits its ability to promote tumor survival [69,80], as well as tumor cell growth and metabolism [62,69].

Although IMiDs can provide therapeutic anticancer, anti-angiogenic, and immunomodulatory effects through cereblon binding, cereblon binding is also implicated in the teratogenic effects of IMiDs [81,82,83] and, additionally, some of the described potentially valuable therapeutic actions of IMiDs in the adult could possibly be devastating to a developing embryo. In relation to teratogenicity, when IMiDs bind to cereblon, the C2H2 zinc finger transcription factor spalt-like transcription factor 4 (SALL4) is ubiquitinated and degraded, which leads to severe birth differences, such as phocomelia [84]. SALL4 is a cereblon neosubstrate important for healthy embryonic limb development and additionally involved in learning and memory [81,84]. SALL4 mutations have been associated with congenital developmental defects, such as Duane–Radial–Ray and Holt–Oram syndromes [27,85,86,87]. As newer research examines the broad pharmacological utility of IMiDs, including possible use for treatment of acute brain injury that is known to occur across gender (including women of child-bearing age), it will be essential to identify novel therapeutic agents that do not produce teratogenicity or other adverse effects. Evading teratogenicity may potentially be achieved with agents that do not act through binding to cereblon and/or do not degrade the neosubstrates Ikaros, Aiolos, and SALL4. 

To this end, it has been found that IMiDs may act through mechanisms independent of cereblon to reduce inflammation. IMiDs reduce the expression of the pro-inflammatory cytokine TNF-α by binding to the 3′ untranslated region (UTR) of TNF-α mRNA and, hence, destabilizing the mRNA, ultimately resulting in reduced translation into protein (Figure 4C) [27,88]. Notable in this regard, phthalimide (Figure 4) and analogs that lack the glutarimide group of thalidomide that is essential to cereblon binding importantly retain TNF-α-lowering actions (see: compounds 24 and 25 in [88]). Given that the mechanism of IMiD action bypasses cereblon binding and neosubstrate degradation, potentially avoiding a known risk of teratogenicity, we have chosen to investigate the cereblon pathway-independent anti-inflammatory properties of IMiDs in the treatment of neuroinflammation and CNS diseases. In the following section, we will introduce FDA-approved IMiDs and detail their potential in treating neurodegenerative conditions, and, thereafter, will discuss new IMiDs.

### 4.2. FDA-Approved IMiDs

In this section, we overview the structure and clinical utilization of current FDA-approved IMiDs.

#### 4.2.1. Thalidomide

Thalidomide (Figure 5A) was first marketed by a German pharmaceutical company in the late 1950s as a sedative alternative to barbiturates and an antiemetic morning sickness drug [28,82,83]. However, it was ultimately taken off the market after it was found to cause severe birth differences [82,83]. The US FDA did not approve thalidomide at the time due to safety concerns of teratogenicity and peripheral neuropathy adverse effects. Years later, in the 1960s, an influential study by Jacob Sheskin, M.D. [89], a physician in Israel, of thalidomide’s unique efficacy in treating erythema nodosum leprosum (ENL), an inflammatory skin condition associated with Hansen’s disease [90], inspired research into other therapeutic benefits of the drug, despite its adverse side effects. ENL is a chronic inflammatory reaction characterized by painful nodules on the skin that can ulcerate and, by systemic involvement, often entails fever and general malaise, as well as other organ effects—and it is considered difficult to treat. Thalidomide’s successful treatment of ENL revived interest, and researchers began to examine other potential therapeutic benefits of the drug. Further, research on additional skin conditions, cancers, and inflammatory conditions revealed that thalidomide was an effective treatment for these disorders [89,91,92] and was a potent inhibitor of TNF-α synthesis [93], thereby initiating research efforts into the development of anti-inflammatory thalidomide analogs [21]. Along with potent anti-inflammatory properties, such agents were also found to possess anti-proliferative, anti-angiogenic, and immunomodulatory actions [94,95], indicating treatment potential for a wide variety of diseases. In 1998, thalidomide was approved by the US FDA for the treatment of ENL and multiple myeloma. 

Given its teratogenicity, it is not surprising that a primary mechanism of action of thalidomide is through cereblon binding and SALL4 degradation [96]. Downstream of cereblon binding and SALL4 degradation, p63 protein abnormalities have also been linked with thalidomide-induced teratogenicity [81,96], and the involvement of other proteins/neosubstrates cannot be ruled out. However, thalidomide also acts through a pathway considered to be cereblon-independent to produce anti-inflammatory effects, via interactions with the 3′ UTR of TNF-α mRNA [21,27,70,88]. Notably, other pharmacological thalidomide actions, such as the antiangiogenic effects, also have a component that appears to be mediated independently of cereblon [63,73,74].

#### 4.2.2. Lenalidomide

Lenalidomide (Figure 5B) is a thalidomide analog that was US FDA-approved to treat multiple myeloma in 2005, an approval which was later expanded to include other blood cancers. The drug was developed as a second-generation thalidomide analog with enhanced potency—in cultured monocytes, lenalidomide inhibited TNF-α secretion with 2000-fold more potency than thalidomide [97,98]. Despite its enhanced potency in the periphery, lenalidomide has been reported to not effectively permeate the BBB, as it acts as a substrate for P-glycoprotein [99], which acts akin to a cerebrovascular pump to expel the drug from the brain [100]. In this light, lenalidomide lacks practicality as a treatment for neuroinflammation, and consequently we have not investigated its analogs in our research.

#### 4.2.3. Pomalidomide

Pomalidomide (Figure 5C), US FDA-approved in 2013 for the treatment of multiple myeloma, is a third-generation thalidomide analog and among the most potent IMiDs available—its TNF-α inhibitory action was reported to be 15,000-fold greater than that of thalidomide in vitro [101]. Additional studies report that pomalidomide suppresses inflammatory factors and inflammation-induced neuronal cell injury in animal models of neurological disease and cellular stress [27]. Importantly, pomalidomide is capable of permeating the BBB and has a brain/plasma ratio of 0.71 [102], making it a potentially viable treatment for neuroinflammation and CNS diseases. We hence focus on developing novel pomalidomide analogs as potential treatments for neuroinflammation and neurodegeneration.

#### 4.2.4. Apremilast

Apremilast (Figure 5D) is a fourth-generation thalidomide analog that does not bind cereblon due to the addition of a large chemical group to thwart the lock-and-key cereblon binding mechanism. Instead, it acts through inhibition of phosphodiesterase (PDE) 4, which modulates various pro-inflammatory signaling molecules and disrupts the inflammatory cascade [103]. Apremilast was US FDA-approved in 2014 to treat moderate to severe plaque psoriasis. Although the drug’s lack of cereblon-associated teratogenicity is advantageous, the compound has poor BBB permeability and thus, like lenalidomide, has poor potential as a treatment for neuroinflammation [21].

### 4.3. New Generation IMiDs

We have focused our studies to consider IMiDs that have potent anti-inflammatory properties (e.g., higher-generation thalidomide analogs), reduced teratogenicity (e.g., evasion of cereblon binding and/or Ikaros, Aiolos, and SALL4 degradation), and CNS accessibility (e.g., adequate BBB penetration) in order to discover novel treatments for neuroinflammation and related neurological disorders (Table 1). Given that, for the purpose of treating neuroinflammation, we do not require anticancer and immunomodulatory properties, there is no consequence to utilizing compounds that do not bind cereblon or degrade neosubstrates related to these actions. Our group has designed analogs around the backbone of pomalidomide to take advantage of its enhanced potency and BBB penetration compared to other IMiDs, and we have also developed a significantly modified thalidomide analog. We have conducted experiments on two of these pomalidomide analogs, 1,6′-dithiopomalidomide (1,6′-DP) and 3,6′-dithiopomalidomide (3,6′-DP), as well as the novel thalidomide-like compound, N-adamantyl phthalimidine (NAP) (Table 1).

We synthesized 1,6′-DP and 3,6′-DP from pomalidomide utilizing Lawesson’s reagent and P_4_S_10_–pyridine complex, respectively [104]. Notably, the P_4_S_10_–pyridine complex method was developed to allow preferential formation of 3,6′-DP in a one-pot procedure [104]. In contrast, NAP was synthesized from a mixture of phthaldialdehyde, 1-(1-adamantyl)ethylamine hydrochloride, and potassium carbonate [28]. These compounds were generated from a substantially larger series of compounds with a goal to generate chemically modified agents to minimize degradation of SALL4, Ikaros, and Aiolos to potentially attenuate teratogenic effects—we thionated the pomalidomide backbone in the dithiopomalidomides to reduce neosubstrate degradation [104] and added steric bulk to NAP to hinder cereblon binding [28]. Our preclinical studies of these three new generation IMiDs in neurodegenerative disease and brain injury show the advantages of using these drugs in several contexts (Table 1).

**Table 1 biomolecules-13-00747-t001:** Chemical structures, cereblon binding, preclinical outcomes, and BBB assessment of new generation IMiDs and of pomalidomide (Pom). 3,6′-DP binds cereblon, but it does not degrade the neosubstrates, whose degradation is associated with teratogenic/anticancer effects (SALL4, Ikaros, and Aiolos). 1,6′-DP also binds cereblon and degrades neosubstrates Ikaros and Aiolos only at very high drug concentrations. NAP does not bind cereblon and, thus, does not degrade these neosubstrates. In preclinical models of TBI, ischemic stroke, AD, and neuroinflammation, each compound has been associated with beneficial effects. Sources: (a.) ref. [105]; (b.) ref. [106]; (c.) ref. [107]; (d.) ref. [108]; (e.) ref. [109]; (f.) ref. [28]. Metrics of BBB permeability are provided by direct measurement of brain and plasma levels to provide a concentration ratio [105], by CNS multiparameter optimization (MPO) score, and by computed Log P (CLog P) value [110]. Finally, all compounds conform to the Lipinski Rule of 5 [111]—aligning with predicted desirable druglike features.

Compound	Cereblon Binding?	Degradation of Neosubstrates?	Preclinical Outcomes	BBB Permeability Evaluation	Lipinski Rule of 5
**3,6′-DP** 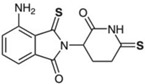	Yes	No	Protects CNS cells in rat TBI models. See (a.) Lin et al., 2020, (b.) Huang et al., 2021, & (c.) Hsueh et al., 2022.	Brain/plasma ratio: 0.8 MPO score: 5.5 CLog P: 0.97	Conforms
			Anti-neuroinflammatory in rat ischemic stroke models. See (d.) Tsai et al., 2022.		
			Anti-inflammatory and neuro- protective in 5xFAD mouse AD model. See (e.) Lecca et al., 2022.		
**1,6′-DP** 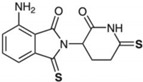	Yes	Yes—Ikaros & Aiolos, at high concentrations only	Mitigates stroke and reduces pro- inflammatory factors in brain. See (d.) Tsai et al., 2022.	MPO score: 5.5 CLog P: 0.97	Conforms
**NAP** 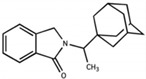	No	No	Improves TBI-driven damage in rat model. See (f.) Hsueh et al., 2021.	MPO score: 3.7 CLog P: 3.86	Conforms
**Pomalidomide** 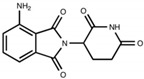	Yes	Yes—SALL4, Ikaros, Aiolos	Mitigates TBI-induced deficits andneuroinflammation at 5-fold higherdose vs. 3,6′-DP. See (a.) Lin et al., 2020 & (b.) Huang et al., 2021.	Brain/plasma ratio: 0.8 MPO score: 4.8 CLog P: −0.16	Conforms

### 4.4. Cereblon Binding and Neosubstrate Degradation of New Generation IMiDs

We performed several screening experiments to evaluate whether 3,6′-DP, 1,6′-DP, and NAP were sufficiently pharmacologically interesting, as compared to thalidomide and pomalidomide, to support both their potential development as candidate drugs and an in-depth evaluation of their tolerability/safety with particular focus on teratogenicity. Our hypothesis was that lack of cereblon binding and/or downstream degradative actions on SALL4 and other neosubstrates would provide a guide to a potentially better-tolerated IMiD. Thionation of the pomalidomide backbone in the synthesis of 3,6′-DP and 1,6′-DP was performed to potentially reduce neosubstrate degradation downstream of cereblon binding. As for NAP, the premise underpinning our design of this compound and similar analogs was to retain the core bicyclic structure of thalidomide-like drugs, but to replace the glutarimide moiety that is key in its binding interactions with cereblon [81,84,112] with an adamantyl moiety, whose steric bulk hinders cereblon binding. We predicted that retaining anti-inflammatory action, but prohibiting cereblon binding and/or neosubstrate degradation, would ensure that any known or unknown proteins potentially ubiquitinated by classical IMiD-cereblon interactions, leading to adverse effects, would be barred from occurring. Such agents could not only provide candidate drugs, but also pharmacological tools to characterize mechanisms underpinning the teratogenicity of classical IMiDs—importantly, whether these are solely mediated through cereblon (as presently largely considered [81,84,112]) or have an independent component.

As indicated in Table 1, we confirmed experimentally that NAP does not bind cereblon (Figure 6A), whereas 3,6′-DP and 1,6′-DP do (Figure 6B). Consequently, NAP does not degrade Aiolos, Ikaros, and SALL4 (Figure 6C) [28]. Of note, despite potently binding cereblon, 3,6′-DP does not lead to degradation of Aiolos and Ikaros [108,109], and 1,6′-DP binding only leads to degradation of Aiolos and Ikaros at high drug concentrations [108]. Remarkably, both dithiopomalidomide drugs maintain expression levels of SALL4 (Figure 6D) [108]. These findings indicate that these drugs may potentially be less teratogenic and safer for patient use than older generation IMiDs, and this supports their further evaluation in this regard. Our collaborative research group has also evaluated the safety profiles of 3,6′-DP and 1,6′-DP with an Ames assay to measure mutagenicity, an in vitro chromosomal aberration assay to measure chromosomal damage, and a human Ether-a-go-go Related Gene (hERG) patch clamp assay to evaluate cardiac safety. For all three assays, essential in the ‘go/no go decision process’ of preclinical drug development, we found no safety issues with 3,6′-DP and 1,6′-DP treatments [108]. Given the ability of NAP to avoid cereblon binding due to its steric bulk and its safety in the above three assays, NAP warrants evaluation of its teratogenicity in classic in vivo models with the knowledge that it retains anti-neuroinflammatory activity.

## 5. 3,6′-DP and 1,6′-DP Experimental Data

This section discusses recent preclinical studies of 3,6-DP that evaluate its efficacy in rodent models of TBI and AD, as well as our comparative study of 3,6′-DP and 1,6′-DP in a rat model of ischemic stroke.

### 5.1. TBI as a Model of Neurodegenerative Disease

Our research group uses TBI rodent models to evaluate the efficacy of promising anti-inflammatory IMiDs. Induced TBI models are versatile preclinical tools because they produce similar pathophysiological sequalae seen in human TBIs, [8], as well as common inflammatory molecules that have been implicated in driving neurodegenerative disease progression in humans [113]. Furthermore, patients diagnosed with TBI early in life have been found to possess an increased risk of developing AD and PD with age [114,115,116,117], emphasizing important links between TBI and chronic neurodegenerative disorders and validating the use of TBI animal models to study mechanisms of and treatments for neurodegenerative disease.

TBI pathology is classified by the primary and secondary phases of the injury. The primary phase occurs immediately after the moment of injury—it includes the mechanical force damage to the brain that can cause neuronal cell necrosis, axonal injury, disruption to the BBB, release of ROS, and the spread of excitatory neurotransmitter release and excessive depolarization [118]. Further, damage to cell membranes from TBI causes rapid sodium and calcium influx into the neuron, resulting in abnormal cell depolarization [119]. Damaged neurons undergoing necrotic cell death release DAMPs into the extracellular space, which can activate cytokine and pattern-recognition receptors, such as purinergic receptors and toll-like receptors (TLRs), which can then initiate intracellular inflammatory signaling [120].

The protracted secondary phase of TBI refers to the changes that time-dependently occur in the hours and days following the initial injury. This phase encompasses processes of neuronal cell dysfunction and death, involving programmed neuronal cell death (PNCD), neuroinflammation, glutamate excitotoxicity, and oxidative stress [121]. Additionally, shortly following TBI, there is a pronounced elevation in the generation and release of pro-inflammatory cytokines, particularly TNF-α, from microglia and astrocytes, along with a change in microglial phenotype [107,122]. Whereas an initial controlled release of TNF-α may valuably initiate homeostatic reparative mechanisms, an excessive and long-term (chronic) release is considered detrimental. In this light, IMiDs that gain brain access to reduce TNF-α protein generation and release and, thereby, lower, rather than totally inhibit the initial spike in TNF-α release, may prove valuable to maintain the time-dependency of TNF-α liberation associated with reparative mechanisms without the detrimental over-excess that triggers degenerative mechanisms. 

In synopsis, to simulate the neuroinflammatory milieu of neurodegenerative diseases, we have utilized rodent TBI models in our studies to evaluate the efficacy of IMiDs—and these are described in the following sections.

### 5.2. Research Findings in TBI Models

#### 5.2.1. 3,6′-DP Mitigates Behavioral Impairments 24 h and Seven Days Post-TBI

Rat TBI models were utilized in our preliminary 3,6′-DP studies (Lin et al., 2020 [105], Huang et al., 2021 [106], and references contained within) before transitioning into additional preclinical models of neurodegeneration. Moderate TBIs were induced in wild-type (WT) animals by using the controlled cortical impact (CCI) technique [105] to generate primary lesions in the brain, which then led to the development of the neuroinflammatory secondary phase of TBI. Our 2020 study with Lin and colleagues [105] examined comparative effects of 3,6′-DP and pomalidomide treatments (administered 5 h post-TBI induction) 24 h following injury induction in rats by use of the CCI TBI model. Subsequent research in 2021 with Huang and colleagues [106] expanded on Lin’s study by adding an additional seven-day follow-up point to ensure that any early favorable efficacy was maintained, rather than lost or resulting in a later detriment.

To investigate the ability of 3,6′-DP to ameliorate cognitive and behavioral deficits associated with TBI-related cognitive decline, Lin and colleagues administered four behavioral assays that tested the animals’ body asymmetry (Elevated Body Swing Test (EBST), Figure 7A), neurological function (modified Neurological Severity Score (mNSS), Figure 7B), sensorimotor function (tactile Adhesive Removal Test (ART), Figure 7C), and motor coordination (beam walk, Figure 7D); these are relatively simple tests to administer with strong face values [123,124]. Notably, 3,6′-DP was found to be more efficacious than pomalidomide at mitigating TBI-induced deficits 24 h post-TBI across all four behavioral assays.

Huang and colleagues subsequently evaluated cognitive outcomes, utilizing two behavioral assays to test general locomotor activity and anxiety-like behavior (Open Field Test, Figure 8A) and hippocampal-mediated memory (Novel Object Recognition Test, Figure 8B). Behavioral testing was conducted both at 24 h and seven days post-CCI induced TBI. These tests validated the beneficial behavioral outcomes of 3,6′-DP treatment found in Lin’s previous study—3,6′-DP was more efficacious than pomalidomide at restoring locomotive deficits and reducing anxiety-related behaviors 24 h post-TBI, and both treatments were comparably efficacious seven days post TBI (Figure 8A). Both 3,6′-DP and pomalidomide restored short-term memory function, as measured 24 h post TBI induction, whereas only 3,6′-DP maintained pre TBI levels at the seven-day mark (Figure 8B). 

Both of our TBI studies illustrated the occurrence of TBI-induced impairments in cognitive and behavioral functions, as well as the ability of 3,6′-DP to mitigate these deficits. Importantly, these investigations demonstrated that IMiD-mediated favorable treatment actions were time-dependently maintained. This is notable in the light of prior studies by McIntosh and colleagues [125], demonstrating that TNF-α knockout (−/−) mice subjected to TBI had a smaller injury lesion and a better behavioral outcome acutely, as compared to similarly TBI-challenged WT mice. However, concerning long-term evaluation, TNF-α knockout (−/−) mice, unlike WT mice, failed to recover and displayed persistent behavioral deficits, thereby demonstrating that TNF-α release following brain injury has both beneficial and adverse consequences. Lowering the time-dependent brain levels of TNF-α, rather than totally inhibiting its release and temporality, might represent a better strategy to pursue as a treatment approach—and this is potentially achievable with a well-designed IMiD.

Given that TBI is known to induce neuroinflammation, our behavioral data raised the question of whether 3,6′-DP may be improving cognition through reducing neuroinflammation, particularly since our drug development studies were predicated on an anti-inflammatory phenotypic screen. To answer this question, we euthanized our animals, following behavioral evaluation and performed immunohistochemical analyses on brain tissue to quantify markers of neuroinflammation and neurodegeneration.

#### 5.2.2. 3,6′-DP Mitigates Microgliosis and Neuroinflammation 24 h and 7 Days Post-TBI

In Lin (2020) and Huang’s (2021) studies, neuroinflammation was evaluated by quantifying pro-inflammatory activated microglia expressing ionized calcium-binding adaptor molecule-1 (Iba1). Upon activation in response to an injurious stimulus, microglia express elevated levels of Iba1 [126], and thus Iba1 is a known marker of pro-inflammatory microglia and, by extension, of neuroinflammation. In Lin and colleagues’ study, postmortem analyses of the cortical contusion area of animals’ brains revealed that 3,6′-DP and pomalidomide treatments reduced TBI-induced elevations in Iba1-positive microglia and mediated the morphological transition of microglia back to the quiescent state at 24 h post-TBI induction. In the subsequent seven-day study by Huang and colleagues, analyses of two hippocampal regions, CA1 and the dentate gyrus (DG), revealed that both 3,6′-DP and pomalidomide significantly reduced Iba1-positive microglia in the CA1 region (Figure 9A,B), but only 3,6′-DP was found to reduce Iba1-positive microglia in the DG.

As an additional marker of neuroinflammation, Lin and colleagues measured mRNA levels and protein expression of pro-inflammatory cytokines TNF-α, IL-6, and IL-1β at 24 h post TBI induction with 3,6′-DP and pomalidomide treatments. Both 3,6′-DP and pomalidomide were found to significantly reduce the TBI-induced elevation of TNF-α, IL-6, and IL-1β protein expression in the cortex of TBI rats. 3,6′-DP notably lowered mRNA levels of these proinflammatory cytokines, as well. In addition, 3,6′-DP was found to, likewise, reduce mRNA expression of caspase-3 in rats and mRNA expression of inducible nitric oxide synthase (iNOS) and cyclooxygenase 2 (COX2) in cultured mouse macrophage cells (RAW 264.7), providing additional evidence for anti-inflammatory and anti-apoptotic properties of 3,6′-DP, and also demonstrating that a five-fold lower dose of 3,6′-DP (0.1 mg/kg body weight) provided similar activity to the higher pomalidomide dose (0.5 mg/kg body weight) evaluated.

#### 5.2.3. 3,6′-DP Outperforms Pomalidomide in Minimizing TBI-Induced Astrogliosis 24 h and Seven Days Post TBI

Reactive astrogliosis is a key component of the neuroinflammatory cycle, and reactive astrocytes release compounds that contribute to a neurotoxic environment [39,45]. Lin and colleagues, in 2020, and Huang and colleagues, in 2021, quantified levels of TBI-induced reactive astrocytes using glial fibrillary acidic protein (GFAP; a known marker of reactive astrocytes), staining with and without pomalidomide and 3,6′-DP treatments in the rats. TBI was found to elevate levels of reactive astrocytes by 1.8-fold, but post-injury administration of 0.1 and 0.5 mg/kg body weight 3,6′-DP reduced these levels by 69.2% and 80.2%, respectively, 24 h post-TBI. Of note, post-injury 0.5 mg/kg body weight pomalidomide treatment did not measurably reduce reactive astrocyte levels 24 h post-TBI. Similar results were observed seven days post-TBI in the CA1 (Figure 9C,D) and DG hippocampal regions, with 3,6′-DP outperforming pomalidomide in reducing astrogliosis.

#### 5.2.4. 3,6′-DP Attenuates Neurodegeneration 24 h and 7 Days Post-TBI

Neuroinflammation, microgliosis, and astrogliosis each contribute to neurodegeneration; we next quantified neurodegeneration itself in TBI animals. Using NeuN as a marker of healthy neurons in immunohistochemical assays, Lin and colleagues found that 3,6′-DP and pomalidomide treatments increased the number of surviving healthy (NeuN-positive) neurons at the cortical contusion site 24 h post TBI, mitigating TBI-induced neuronal loss. In our subsequent TBI study with Huang and colleagues, we utilized Fluor-Jade C (FJC; a marker of degenerating cells) dye and found that 3,6′-DP and pomalidomide treatments significantly reduced the number of degenerating (FJC-positive) cells with neuron-like morphology in the CA1 (Figure 9E,F) and DG hippocampal regions, with 3,6′-DP being more efficacious than pomalidomide in the DG. Considering that 3,6′-DP was also found to be more efficacious than pomalidomide in attenuating microgliosis and astrogliosis, the enhanced efficacy of 3,6′-DP in reducing neurodegeneration is consistent with these findings and highlights a potential link between neuroinflammation and neurodegeneration. Combined with our behavioral results, these data suggest that 3,6′-DP may have a uniquely enhanced capability to reduce neuroinflammation-driven neurodegeneration, and, thus, it warrants further evaluation as a candidate drug to mitigate neurological disorders in which a neuroinflammatory component is prevalent. 

#### 5.2.5. F-3,6′-DP

In a subsequent study we conducted with Hsueh and colleagues in 2022 ([107] and references contained within), we investigated a fluorinated dithiopomalidomide, F-3,6′-DP, intended to prolong metabolism while evading teratogenic/anticancer mechanisms associated with SALL4, Aiolos, and Ikaros degradation. Fluorine molecules are both small in size (F van der Waals radius: 1.47 Å, versus H: 1.20 Å) and strongly electronegative (and, hence, have a high electron withdrawing property) and can thus have a profound effect when used in a number of ways to replace a hydrogen atom in the medicinal chemistry of small organic biologically active compounds. In particular, substitution with a fluorine at or close to a position of phase 1 (cytochrome P450) metabolic attack can potentially slow metabolism and, thereby, help maintain therapeutic drug levels [127,128], as the C-F bond is relatively more resistant to attack than is the C–H bond. Likewise, in specific cases, the introduction of a fluorine atom can favorably alter the pK_a_ of the resulting compound, and thus improve its bioavailability. However, the same relative sizes of fluorine and hydrogen and their similar bond lengths, when connected to a carbon, allow substitution without substantially altering the resulting size of the molecule and, in general, permit it to interact and bind with its target protein in a manner similar to that of the original non-fluorinated compound. Additionally, potentially valuable, inclusion of a fluorine can, in specific cases, support imaging of the compound, as with ^18^F positron emission tomography (PET) and ^19^F magnetic resonance imaging (MRI). 

We confirmed in cultured human multiple myeloma (MM1.S) cells that F-3,6′-DP did bind cereblon but did not degrade Aiolos, Ikaros, and SALL4, and we found numerous anti-inflammatory benefits of the drug in TBI-induced rats. Behavioral evaluation of motor coordination revealed a marked improvement in TBI-related motor deficits with F-3,6′-DP treatment, and the compound reduced reactive astrogliosis and microglial activation in the animals’ brains. Additionally, F-3,6′-DP reduced expression of pro-inflammatory cytokines, including TNF-α, in LPS-induced neuroinflammation models in rats and in cultured mouse macrophage (RAW 264.7) cells.

### 5.3. 3,6′-DP and 1,6′-DP as Ischemic Stroke Treatments 

For a compound demonstrating favorable anti-inflammatory action and efficacy in one neuroinflammatory disorder, such as TBI, such activity should theoretically cross over to other disorders with a prevalent inflammatory element. Evaluating whether or not this occurs can de-risk the drug development scheme. In this regard, having demonstrated that 3,6′-DP mitigates TBI-induced neuronal cell loss, cell models of hypoxia suggest that the anti-inflammatory capability of 3,6′-DP may be effective in mitigating hypoxia-induced neuronal death [129]. Cerebral ischemia is a common complication after initial injury in TBI and can last for up to a week after mild to moderate TBI, exacerbating damage and resulting in neural dysfunction and death [130,131,132]. Our 2022 study with Tsai and colleagues [108] explored this potential to treat stroke with new generation IMiDs and compared the efficacy of pomalidomide, 3,6′-DP, and 1,6′-DP on rats with transient induced middle cerebral artery occlusion (MCAo). 

Tsai and colleagues demonstrated that pomalidomide, 3,6′-DP, and 1,6′-DP improved behavioral consequences, histological damage, and inflammation resulting from stroke ([108] and references contained within). Infarct volumes (Figure 10A) and body asymmetry (measured by the EBST) (Figure 10B) were reduced in all three treatment groups when compared with vehicle-treated MCAo animals. We additionally evaluated levels of pro-inflammatory cytokines IL-1β and TNF-α and anti-inflammatory cytokine IL-10 to determine whether pomalidomide, 3,6′-DP, and 1,6′-DP may alter the ratio of pro-inflammatory to anti-inflammatory cytokines and thus mitigate inflammation. These cytokine levels were quantified in plasma samples from the animals 24 h post MCAo. Across all treatments, levels of pro-inflammatory cytokines IL-1β and TNF-α were reduced relative to the untreated MCAo condition, with 3,6′-DP and 1,6′-DP reducing IL-1β levels to a greater extent than did pomalidomide (Table 2). Interestingly, only the 3,6′-DP treatment was effective in significantly increasing levels of anti-inflammatory cytokine IL-10 (Table 2). This finding could indicate that all three pomalidomide analogs exert anti-inflammatory effects through the reduction of pro-inflammatory cytokine levels, but only 3,6′-DP provides further anti-inflammatory action by additionally increasing anti-inflammatory cytokine levels. An additional potential pharmacological action of all three compounds evaluated is to ameliorate BBB damage and resulting elevated permeability present in brain, following ischemic stroke, that is also evident after a TBI. In a recent human and rodent study of radiation-induced BBB injury, thalidomide was found to mitigate pericyte damage and, thereby, stabilize the neurovasculature and BBB function by restoring the expression of platelet-derived growth factor receptor β in radiation-damaged pericytes [133]. In this regard, our prior studies demonstrated that the IMiD 3,6′-DP mitigated disruption of the BBB and infiltration of systemic leukocytes, following MCAo in mice [134], thereby adding to the anti-neuroinflammatory effects of this drug class. 

As observed in our previous TBI studies, 3,6′-DP provided substantial anti-inflammatory activity in the rodent MCAo model of ischemic stroke and, as it demonstrates much potential as a treatment for neuroinflammation, our research team consequently examined its effects in a further model of neurodegeneration—in a transgenic mouse model of AD. 

### 5.4. 3,6′-DP Treatment in an AD Mouse Model

Given our prior research findings that 3,6′-DP mitigates neuroinflammation and further neurodegenerative consequences of TBI and stroke, our research group also investigated the treatment effects of this compound in the hallmark of neurodegenerative disorders, AD. Neuroinflammation is a key characteristic of the AD brain; yet, current treatments, including the recent US FDA-approved anti-amyloid-β monoclonal antibody drugs Aducanumab and Lecanemab, target amyloid-β rather than neuroinflammatory pathways. Specifically, Aducanumab and Lecanemab have been reported to clear the brains of AD patients of amyloid-β plaques. However, clinical trial data from the two identically designed Aducanumab 18-month randomized, double-blind, placebo-controlled, parallel-group studies (involving 3285 participants at 348 sites across 20 countries) that evaluated the efficacy, safety, pharmacokinetic, and pharmacodynamic properties, have raised concerns as to whether statistically significant outcomes in relation to amyloid-β biomarkers provided any clinically meaningful effects for the drug [135,136,137]. Hence, whether Aducanumab has clinically useful effects on cognition in AD remains controversial. In much the same manner, the Lecanemab phase III Clarity AD clinical trial that supported the agent’s recent US FDA approval (6 January 2023) demonstrated statistically favorable changes in amyloid-β biomarkers and, additionally, statistically met its primary outcome measure (a statistical difference from placebo as per the Clinical Dementia Rating–Sum of Boxes (CDR-SB) score), together with select secondary measures that likewise relate to cognition. Although statistical significance was met, it remains open as to how much Lecanemab meaningfully impacts cognition in AD, as its achieved treatment benefit of 0.45 points on the 18-point CDR-SB score is well below the minimal clinically important difference in AD clinical trials [138]. In this light, although Aducanumab and Lecanemab can be considered as advances in AD treatment, it is clear that much more is required for a treatment to be truly considered effective. These and other studies have opened questions as to why substantial reductions in various forms of amyloid-β do not appear to dramatically impact loss of cognition in AD. Genetic studies clearly point to the involvement of elevated and awry processing of amyloid-β in the development of AD, but what additional factors potentially triggered by amyloid-β more directly drive cognitive loss (whether hyperphosphorylated tau, neuroinflammation or, indeed, something else) remain open and could provide a valuable drug target. 

In our study by Lecca and colleagues in 2022 [109], we employed 3,6′-DP and pomalidomide as tools to shed light on the effects of neuroinflammation on important deficits common to AD, such as synaptic dysfunction and cognitive decline. As a model of AD, we utilized 5xFAD mice and treated with pomalidomide or 3,6′-DP over a four-month period. We hypothesized that, if 3,6′-DP and pomalidomide are effective in improving cognitive ability, then we should observe improvements in markers of neuroinflammation and gliosis, as well, providing evidence for neuroinflammation and gliosis as drivers of AD.

#### 5.4.1. 3,6′-DP Treatment Mitigates Cognitive Decline in 5xFAD Mice

In 5xFAD mice, Lecca and colleagues ([109] and references contained within) assessed cognitive functions including spatial working memory, working memory, stress reactivity, anxiety, and motor function using two behavioral paradigms: a Y-maze test and an open field test. During the Y-maze test, we quantified the animals’ alternation rates to evaluate spatial working memory. This test revealed that four-month 3,6′-DP and pomalidomide treatments increased spatial working memory function to near WT levels, with 3,6′-DP trending higher than pomalidomide. In the open field test, we quantified the percentage of the total distance that the animals traveled that was in the center of the field to evaluate anxiety-like behaviors, as well as latency to visit each of the four corners of the field to evaluate working memory, stress reactivity, and motor function. Four-month 3,6′-DP treatment again demonstrated enhanced cognitive benefits, as we observed reduced anxiety-like behaviors (reduced percent of distance in the center of the field) and elevated working memory and motor function (reduced latency to visit all corners) in the 3,6′-DP-treated animals. 

To summarize and increase the sensitivity of these behavioral data, we calculated a composite behavioral score (Figure 11). Of note, 5xFAD mice had significant cognitive impairments relative to WT, but four-month 3,6′-DP treatment prevented the development of these cognitive deficits in 5xFAD mice, with a composite behavioral score near the WT score. Interestingly, the same preventive effect was not observed with pomalidomide treatment. These behavioral data yield encouraging results on 3,6′-DP as a potential treatment that could preemptively ameliorate cognitive consequences of AD.

#### 5.4.2. 3,6′-DP Mitigates Neuroinflammation in 5xFAD Mice

To determine the effects of 3,6′-DP and pomalidomide on mitigating neuroinflammation, we measured microglial activation and astrogliosis in two brain regions with relevance to AD: the hippocampus is responsible for memory formation, and the cortex is, likewise, responsible for multiple cognitive functions. Using the same assays as in our TBI studies, we quantified Iba1 immunoreactivity in the hippocampus (Figure 12A,B) and cortex (not shown) of 5xFAD mice. In these assays, 3,6′-DP attenuated microglial activation in both regions, whereas pomalidomide appeared to mitigate activation, but not at statistically significant levels. Further, we assessed the phenotypic state of microglia by examining microglial morphology across the different treatment groups and found that 3,6′-DP and pomalidomide treatments largely returned microglia to their quiescent/surveillant morphology. To quantify reactive astrogliosis, we assayed for GFAP-positive cells and found that both pomalidomide and 3,6′-DP treatments significantly reduce reactive astrocyte levels in hippocampal (Figure 12C,D) and cortical regions (not shown) of 5xFAD mice. The observed cognitive benefits of our treatments, hence, are associated with a marked attenuation of neuroinflammation, as indicated by reduced levels of activated microglia and reactive astrocytes.

#### 5.4.3. 3,6′-DP Mitigates Synaptic Loss and Neurodegeneration in 5xFAD Mice

Neuroinflammation can promote loss of synapses, as TNF-α is involved in synaptic plasticity, synaptic scaling, and cleavage [139,140,141,142], and thus aberrant levels of TNF-α can compromise synaptic integrity. Synaptic loss is a key feature of AD and is a primary contributor to AD-associated cognitive decline [143]. Thus, we next measured markers of synaptic loss and neurodegeneration to complete the hypothesized connection between neuroinflammation, cognitive decline, and neurodegeneration. We measured expression of postsynaptic density protein 95 (PSD-95) as a marker of intact synapses in the hippocampus (Figure 12E,F) and cortex (not shown) to evaluate whether the behavioral improvements we observed in 3,6′-DP-treated 5xFAD mice may be associated with improvements in synaptic integrity. We found that 3,6′-DP and pomalidomide treatments completely reversed the loss of PSD-95-expressing dendritic spines in the hippocampus and cortex that was observed in vehicle-treated 5xFAD mice. 

In addition, we labeled degenerating neuronal cells with FJC dye to quantify neurodegeneration in the hippocampus (Figure 12G,H) and cortex (not shown) of 5xFAD mice. We found that 3,6′-DP and pomalidomide treatments mitigated the number of degrading neurons seen in vehicle-treated 5xFAD mice by approximately 20%, with pomalidomide reaching a statistically significant difference from vehicle-treated 5xFAD mice in the hippocampus. 

#### 5.4.4. Independence of Treatment Effects from Amyloid-β

Amyloid-β plaque formation, as noted, is a well-known hallmark of AD, and thus currently approved treatments for AD largely target amyloid-β generation pathways or are focused to efficiently clear amyloid-β forms. However, as discussed, these treatments have not proven to be adequate in remediating AD cognitive loss. We have posited that an anti-neuroinflammatory approach may be a more effective treatment strategy. Rather than being an epiphenomenon and inert bystander of the disease process, neuroinflammation may provide an essential link between awry amyloid-β generation and the cognitive loss that ultimately occurs. In order to validate this hypothesis, we tested the effects of pomalidomide and 3,6′-DP on amyloid-β levels. We hypothesized that, if we do not see an effect of our compounds on amyloid-β, then we can attribute the cognitive benefits observed with our treatments to their anti-neuroinflammatory properties and not to amyloid-β mitigation. We can extend this hypothesis to suggest that neurodegenerative pathology may be primarily driven by neuroinflammation, and appropriate neuroinflammation-targeting treatments, such as achieved by a safer IMiD, may be more efficacious and worth adding to the currently limited armamentarium of available drugs for AD. 

Elevated levels of amyloid-β have been identified in the brains of 5xFAD mice [144], and amyloid-β plaque deposition was observed in the hippocampus and cerebral cortex of our 5xFAD mice at eight months. To investigate the effects of 3,6′-DP and pomalidomide on amyloid-β, we measured amyloid-β plaque load and levels of soluble and insoluble amyloid-β 40 and 42 at eight months in all treatment groups. We found that neither 3,6′-DP, nor pomalidomide, significantly altered plaque load, nor concentrations of soluble and insoluble amyloid-β 40 and 42, in the brains of 5xFAD mice. We also did not observe significant differences in the ratio of amyloid-β 42/40 peptides between the treatment groups. These data support our hypothesis that the behavioral improvements observed with 3,6′-DP and pomalidomide treatments occur independently from the amyloid secretory pathway and are primarily a result of the mitigation of neuroinflammation. These data point to neuroinflammation (initially induced by over-generation of amyloid-β species in the 5xFAD mouse) as a key driver of synaptic dysfunction and loss, neurodegeneration, and cognitive decline, and they additionally support 3,6′-DP and analogs as candidate drugs for AD, which warrant further evaluation. 

### 5.5. 3,6′-DP Experimental Data Summary

Our research group’s work in several different preclinical models of CNS diseases—TBI, ischemic stroke, and AD—has revealed repeatable neuroprotective benefits of 3,6′-DP. This new generation IMiD has demonstrated efficacy in mitigating neuroinflammation, microgliosis, astrogliosis, neurodegeneration, and cognitive impairments in each model system, and its actions were proven to be independent of amyloid-β activity in the 5xFAD model. In this light, 3,6′-DP and close analogs that bind cereblon but do not trigger downstream actions of SALL4 and other proteins warrant further evaluation as drug candidates. Recent evaluations find that 3,6′-DP meets fundamental US FDA requirements for future consideration as a potential human drug candidate in relation to lack of genotoxicity liability by the classical bacterial reverse mutation (Ames) assay to appraise mutagenicity, as well as by the in vitro chromosomal aberration assay to assess chromosomal damage [108]. Additionally, cardiac safety by the hERG patch clamp assay was established [108]. In future studies, we plan to develop 3,6′-DP or a close analog as a potential clinical treatment for neurodegeneration and continue to examine the role of neuroinflammation in neurodegenerative disease progression. 

## 6. NAP Experimental Data

In this section, we discuss our recent preclinical studies investigating the atypical IMiD, NAP, as a treatment for neuroinflammation, neurodegeneration, and cognitive impairments in a mouse model of TBI and a cell culture model of neuroinflammation. We will also overview preclinical data on an additional novel compound—tetrafluorobornylphthalimide—with similar chemical characteristics to NAP in reducing TBI-induced neuroinflammation.

### 6.1. NAP as a Treatment for Neuroinflammation in Preclinical Models

Prior to in vivo TBI studies, Hsueh and colleagues in 2021 ([28] and references contained within) evaluated the anti-inflammatory properties of NAP in mouse macrophage cells (RAW 264.7) and rats challenged with lipopolysaccharide (LPS) and in primary neurons challenged with α-synuclein and amyloid-β. This work allowed us to compare the anti-inflammatory mechanisms of NAP to that of our other pomalidomide agents and across phenotypic models of neuroinflammation. In cultured RAW 264.7 cells, NAP mitigated TNF-α levels, indicating reduced inflammation. In primary neuronal cultures, NAP significantly reduced neuronal cell death induced by α-synuclein and amyloid-β, and microglial activation induced by these misfolded proteins. In rats challenged with LPS, we found that NAP lowered LPS-induced heightened levels of IL-6 in the brain and of TNF-α in the brain and plasma, with, importantly, no change to levels of anti-inflammatory cytokines IL-10 and IL-13. These findings suggest that, despite lacking cereblon binding, NAP exerts anti-inflammatory effects in a similar manner to 3,6′-DP, 1,6′-DP, and pomalidomide, by reducing levels of pro-inflammatory cytokines and consequently promoting a neuroprotective environment. 

### 6.2. NAP as a Treatment for TBI-Induced Neuroinflammation and Neurodegeneration

In 2021, Hsueh and colleagues published an in vivo preclinical study exploring the effects of NAP treatment in TBI. In this study, we evaluated the efficacy of NAP in mice 2 weeks post- CCI-induced TBI. This research, as emphasized previously, provides us with a tool to assess the impact of neuroinflammation on neurodegenerative disease progression and consequent cognitive impairment. As in the previous section, we will discuss the effects of NAP treatment on cognitive outcomes, neuroinflammation, and neurodegeneration in this animal model of TBI and explore the interplay between these factors. 

#### 6.2.1. NAP Improves Behavioral Impairments Post-TBI

Based on our preliminary results in cell culture and rats, Hsueh and colleagues sought to determine whether the evident anti-inflammatory properties of NAP influenced behavioral outcomes post-TBI. We administered NAP to mice 5 h and 24 h, following TBI induction, and we performed behavioral assays one and two weeks later, comparing the data to pre TBI results. We utilized the EBST to evaluate motor impairments (Figure 13A), a tactile ART paradigm to evaluate somatosensory function (Figure 13B), and a beam walking test (BWT) to evaluate motor coordination (Figure 13C). Notably, NAP appeared to improve behavioral outcomes across the three tests and, interestingly, the behavioral results demonstrate that, similar to humans, a spontaneous recovery occurs in TBI+saline-treated animals consequent to homeostatic adaptive and reparative mechanisms—and these appear to be accelerated (i.e., occurring earlier) in NAP-treated TBI animals. For the EBST, NAP significantly reduced contralateral turns at one-week post TBI when compared to TBI saline mice, with the difference no longer evident at two weeks post-TBI due to spontaneous recovery in the TBI+saline animals. For the ART paradigm, at one-week post TBI, there was no significant difference between the NAP- and saline-treated mice, but, at two weeks, NAP-treated TBI mice spent a significantly shorter time removing stickers from their contralateral paw. For the BWT, NAP significantly reduced motor deficits one-week post TBI, and this was no longer evident at two weeks, consequent to spontaneous recovery in the TBI+saline group. These data suggest a time-dependency to TBI-induced behavioral impairments and a consistent ability of NAP treatment to accelerate recovery, supportive of further study of NAP. 

#### 6.2.2. NAP Mitigates Neuroinflammation Post TBI

We sought to examine whether the observed behavioral improvements with NAP treatment were associated with reduced neuroinflammation (this section) and improved synaptic health (next section). To assess severity of neuroinflammation, we measured microgliosis and determined microglial phenotypic changes by quantifying Iba1-positive immunoreactivity and imaging microglial morphology, as in our prior experiments with the dithiopomalidomides. We observed that, following injury, microglia swelled and reduced the size of their processes in transitioning into a more active pro-inflammatory state. Although microglia exist across a broad spectrum of phenotypes, we classified microglial morphology into four types and two activation states for our purposes: ramified and intermediate for the resting/surveillant state and ameboid and round for the activated/pro-inflammatory state (Figure 14A). As a result of NAP treatment, we observed a reduced number of activated microglia and reduced total number of microglia in the ipsilateral (injured) thalamus post-TBI, reversing an increase that was observed in the saline-treated TBI group (Figure 14B).

As an additional measure of neuroinflammation, Hsueh and colleagues evaluated the effects of NAP treatment on pro-inflammatory cytokine TNF-α levels and found that NAP reduced the number of TNF-α-positive and TNF-α/Iba1-positive dual-labeled cells in the cortex of TBI mice. This finding suggests that, in addition to altering microgliosis post-TBI, NAP also mitigates inflammation by reducing the production of TNF-α in microglial cells.

#### 6.2.3. NAP Mitigates Synaptic Loss Post-TBI

We additionally investigated the effects of NAP in mitigating synaptic loss in our TBI mice. To do so, we measured expression of PSD-95, a postsynaptic marker, in the CA1 region of the hippocampus. We found that, after TBI, PSD-95 expression was substantially reduced in the ipsilateral (injured) CA1 region, reflective of a severe loss of synapses. Notably, NAP treatment reversed this deficit, restoring levels of PSD-95. NAP thus demonstrates a capability to restore synaptic health in the injured brain, an encouraging characteristic for future neurodegenerative disease drugs.

### 6.3. A Newly Developed Non-Cereblon-Binding IMiD in Initial Phases of Study: Tetrafluorobornylphthalimide

Our collaborative research group has very recently synthesized and initiated study of a further novel non-cereblon-binding ImiD with a likely similar mechanism of action as NAP. Tetrafluorobornylphthalimide (TFBP) retains the phthalimide moiety of ImiD compounds, but the glutarimide ring was replaced by a bridge structure to inhibit cereblon binding, a similar concept to the steric bulk, preventing NAP from binding cereblon. Extensive fluorination was added to potentially mitigate phase 1 metabolic processes. As reported by Lecca and colleagues in 2023 ([145] and references contained within), after confirming minimal cereblon binding and a lack of neosubstrate (SALL4) degradation in cell culture, TFBP was evaluated in several preclinical models of the neurodegenerative milieu, including LPS-challenged RAW 264.7 cells, LPS-challenged rats, and CCI TBI mice. In the LPS-challenged cells and rats, we found that TFBP reduced proinflammatory cytokine generation. In TBI challenged mice, we observed a mitigation of TBI-induced neuroinflammation and an expedited recovery of behavioral abilities in TFBP-treated animals relative to controls. In classical chicken embryo studies, TFBP lacked evident teratogenic actions, and it will continue to be investigated by our research group as a promising atypical IMiD that retains anti-inflammatory actions in the absence of cereblon binding.

### 6.4. NAP and TFBP Experimental Data Summary

Our studies of NAP and TFBP treatments in TBI models of neurodegeneration have provided evidence for the compounds’ abilities to ameliorate behavioral deficits related to CNS injury, reduce neuroinflammation and microgliosis, and promote synaptic health. Importantly, NAP and TFBP evade cereblon binding, and the latter appears in preliminary in vivo studies to not produce obvious teratogenic effects. This new generation of atypical IMiDs demonstrate much potential for offering anti-inflammatory and neuroprotective benefits with reduced risk of key adverse effects. As discussed earlier and within the recent TFBP publication [145], anti-inflammatory actions can be mediated via cereblon-dependent and independent pathways. In line with this, cereblon overexpression leads to suppressed NF-*κ*B activation and lower pro-inflammatory cytokine levels in response to TLR4 stimulation with LPS. In contrast, cereblon knockdown results in an amplified pro-inflammatory response [146]. Classical thalidomide-like drugs can, hence, potentially induce anti-inflammatory actions mediated through cereblon E3 ubiquitin ligase-dependent and independent mechanisms. The ability of NAP and TFBP to sufficiently do so in the absence of human cereblon binding warrants further evaluation of these compounds as potential drug candidates for clinical translation.

## 7. Conclusions

Our collaborative research group’s preclinical studies demonstrate that our new generation IMiDs mediate neuroinflammation through numerous mechanisms, thereby highlighting their clinical relevance for slowing neurodegenerative disease progression and improving cognitive and behavioral outcomes. We have additionally made use of these agents as pharmacological tools to explore the role of neuroinflammation in the progression of neurodegenerative disease [109]. Our findings support our hypothesis that neuroinflammation is a primary driver of neurodegeneration, and treatments targeting neuroinflammation and cytokine (particularly TNF-α) synthesis, such as IMiDs or CSAIDs, may be more effective in improving cognitive outcomes than traditional amyloid-β targeting treatments or classical anti-inflammatory NSAIDs. Current treatments for TBI and neurodegenerative disorders fail to slow disease progression and, instead, largely treat symptoms or clear amyloid-β without substantially impacting cognitive outcome measures. The compounding cycle of neuroinflammation and neurodegeneration can be broken with the use of anti-neuroinflammatory and neuroprotective compounds, such as with new generation IMiDs [109]. Classical IMiDs, such as thalidomide, are encumbered by adverse actions, in particular teratogenicity, and although the typical AD and PD patient populations are elderly and not within childbearing years, TBI and associated disorders impact all ages. Moreover, Hansen’s disease remains a largely neglected tropical disease that still occurs in more than 120 countries, with an excess of 200,000 new cases reported every year [147]. Thalidomide remains the treatment drug of choice and, despite its known teratogenicity, is prescribed widely, resulting in a new generation of thalidomide birth differences [148]. Hence, a new potentially safer analog would be valuable. In this regard, atypical and classical IMiDs that either do not bind cereblon or bind it, but do not trigger degradation of downstream transcription factors, such as SALL4, Ikaros, and Aiolos, known to be involved in the teratogenic, anticancer, and anti-angiogenic actions of thalidomide-like drugs, may represent a strategic approach to gaining safer IMiDs. Whereas, the teratogenicity of thalidomide-like drugs appears to be driven by cereblon-triggered degradation of SALL4 and associated transcription factors [69,83,84,112], and the classical adverse actions of clinically approved IMiDs, when used in cancer treatment (anemia, neutropenia, thrombocytopenia [149]), potentially derive from their anticancer actions mediated via cereblon triggered Ikaros/Aiolos degradation, and may or may not occur in the described new IMiDs that do not induce such degradation. 

It should be cautioned, however, that, whereas, the anti-inflammatory actions of IMiDs clearly involve cereblon-dependent and -independent mechanisms, so too may pathways that underpin teratogenic, anticancer, and anti-angiogenic actions. Indeed, cereblon-independent anti-angiogenic IMiD actions have been reported [73], and the hypnotic effects of thalidomide likewise appear to be independent of cereblon [150]. Other proteins, in addition to SALL4, such as the tumor protein p63 that has multiple isoforms, the major of which are TAP63 and ΔNP63, are subject to thalidomide-mediated degradation and appear to be involved in different teratogenic phenotypes [151]. Likewise, PLZF is a further neosubstrate potentially involved in the teratogenicity of thalidomide and analogs, with its knockdown causing limb differences in chicken embryos [152]. Although these neosubstrates are reported to be degraded via cereblon E3 ubiquitin ligase-dependent action [151,152,153], the possibility of a cereblon-independent mechanism nevertheless exists, as does the potential involvement of yet to be characterized proteins. Finally, it should be borne in mind that the same anti-angiogenesis and anticancer actions of thalidomide and clinical analogs that have proved so valuable in the treatment of adults with cancer might prove to be devastating to a developing embryo, and that thalidomide and lenalidomide, in contrast to pomalidomide, have been reported to induce neurotoxicity [154]. 

The use of non-cereblon binding NAP and TFBP as pharmacological tools could potentially aid in identifying whether proteins such as p63 isoforms and PLZF are degraded by classical IMiDs via solely cereblon-dependent mechanisms. Additionally, as our novel IMiDs provide neuroprotective benefits and work through multiple mechanisms to mitigate neurodegeneration, they thus warrant further study (in relation to both their potentially valuable and adverse actions) as potential therapies for neurodegenerative disease, as well as for systemic disorders with an inflammatory component. In closure, this article reviews a series of new classical and non-classical IMiDs generated on the backbone of pomalidomide and thalidomide to specifically tune their use to mitigate neuroinflammation (a common feature across acute and chronic neurodegenerative disorders [21] and, quite possibly, several neuropsychiatric disorders [27]). Long-term toxicology and in-depth teratogenicity studies are clearly required before use of these new IMiDs can be considered in humans (looking for toxicities known to be associated with classical clinically approved IMiDs [82,83], as well as new potential adverse effects), together with studies to evaluate possible changes in homeostatic mechanisms relating to pro-inflammatory signaling pathways to evaluate whether the anti-inflammatory promise that these new IMiDs offer can be maintained over time.

## Figures and Tables

**Figure 1 biomolecules-13-00747-f001:**
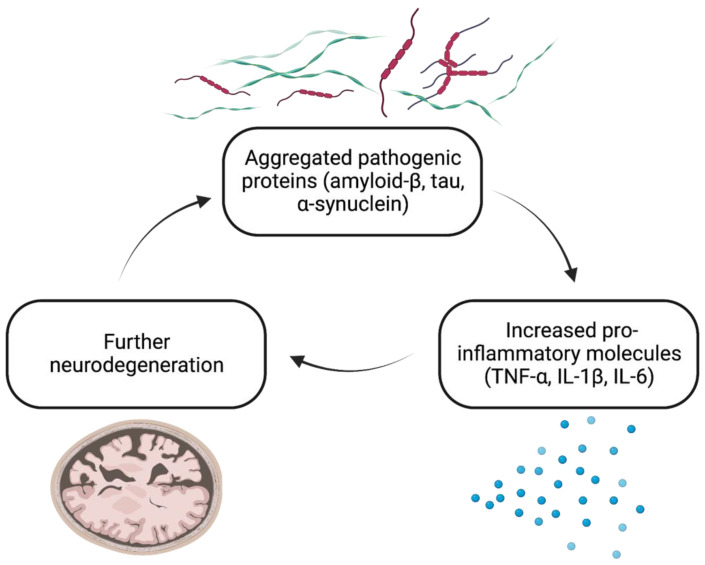
Cycle of neuroinflammation and neurodegeneration. The accumulation of aggregated pathogenic proteins, often in a neurodegenerative disease state, stimulates excessive generation and release of pro-inflammatory molecules, including cytokines, such as TNF-α, IL-1β, and IL-6. These pro-inflammatory molecules promote neuroinflammation and neurodegeneration. Chronically elevated levels of these cytokines and chemokines promote further degradation of tissue and functional abnormalities associated with neurodegeneration, which leads to more release and a feed-forward loop. Created with BioRender.com (accessed on 17 February 2023).

**Figure 2 biomolecules-13-00747-f002:**
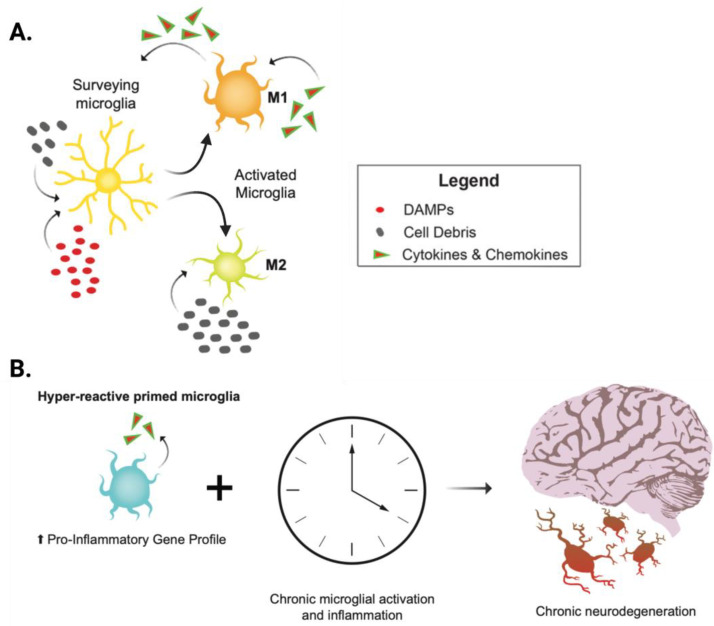
Microglial activation and neuroinflammation. (**A**) In a quiescent state, microglia survey their environment for insults and injury in the CNS. Once they detect damage signals, such as DAMPs, cell debris, cytokines, and chemokines, microglia transition into their M1 activated state. M1 microglia are pro-inflammatory and release cytokines and chemokines that serve to activate other cells. In contrast, M2 microglia have anti-inflammatory and phagocytic functions, clearing cell debris. (**B**) Chronically activated microglia exist in a pro-inflammatory, hyper-reactive primed state and demonstrate enhanced sensitivity to inflammatory stimuli and, thereby, contribute to chronic neuroinflammation and neurodegeneration. Image adapted from Glotfelty et al., 2019 [8].

**Figure 3 biomolecules-13-00747-f003:**
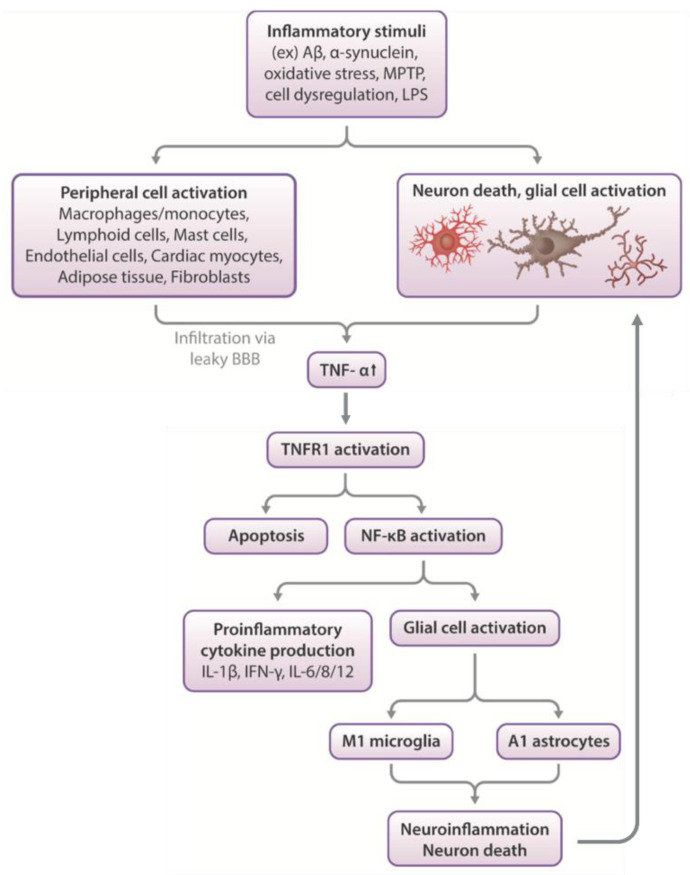
Positive feedback loop of TNF-α-promoted neuroinflammation. Inflammatory stimuli, often associated with the pathophysiological characteristics of a neurodegenerative disease, are neurotoxic and activate peripheral and glial cells. Activation of peripheral and glial cells increases levels of TNF-α, which binds and activates its receptor TNFR1. TNFR1 signaling stimulates apoptosis and activates NF-κB, a transcription factor that enhances the production of pro-inflammatory cytokines and further activates glial cells, causing microglia and astrocytes to adopt their reactive states. Activated microglia and astrocytes contribute to the neuroinflammatory environment and neuronal death, stimuli which promote additional glial cell activation, recommencing the cycle. Image adapted from Jung et al., 2019 [21].

**Figure 4 biomolecules-13-00747-f004:**
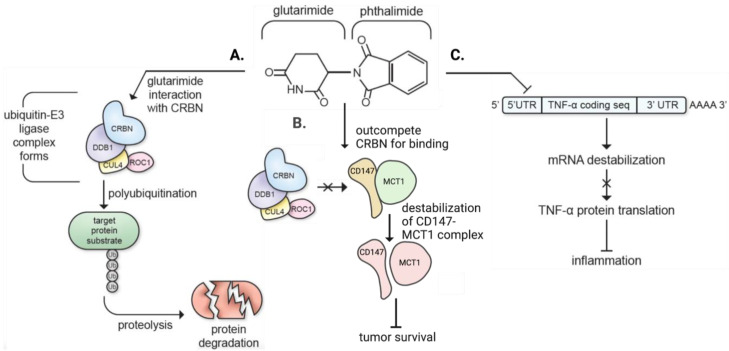
Molecular mechanisms of IMiD action. (**A**) An IMiD compound interacts with cereblon (CRBN) via its glutarimide moiety, altering the substrate preferences of the ubiquitin-E3 ligase complex and leading to the degradation of specific target proteins. (**B**) IMiDs compete with CRBN for binding to the CD147-MCT1 complex, destabilizing the complex and inhibiting its ability to enhance tumor survival. (**C**) An IMiD compound binds to the 3′ untranslated region (UTR) of TNF-α mRNA, serving to destabilize the mRNA and reduce protein expression of TNF-α. Image adapted from Jung et al., 2021 [27]. Created with BioRender.com.

**Figure 5 biomolecules-13-00747-f005:**
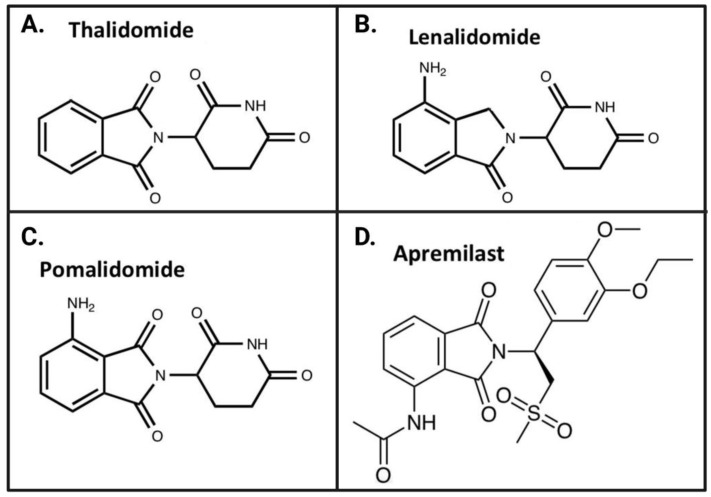
Chemical structures of US FDA-approved IMiDs: thalidomide (**A**) and analogs lenalidomide (**B**), pomalidomide (**C**), and apremilast (**D**). Image adapted from Jung et al., 2021 [27].

**Figure 6 biomolecules-13-00747-f006:**
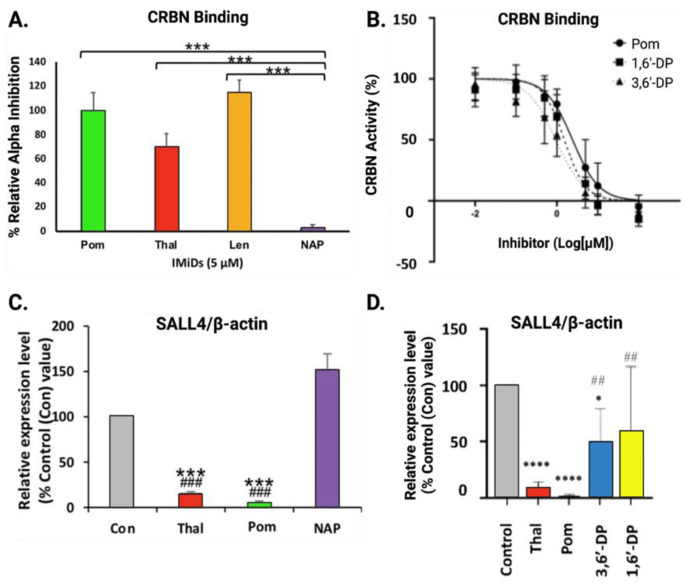
Cereblon binding and reduced SALL4 degradation of NAP, 3,6′-DP, and 1,6′-DP, as compared to older generation IMiDs. (**A**) Binding to cereblon was quantified utilizing a cereblon/BRD3 binding FRET assay. When 5 μM pomalidomide (Pom), thalidomide (Thal), lenalidomide (Len), and NAP were tested and compared to a vehicle control (Con), which was set at 0% cereblon inhibition, NAP was not statistically different from the control value, indicating that NAP does not bind cereblon. *** *p* < 0.001 versus control value. (**B**) 3,6′-DP and 1,6′-DP bind cereblon with similar potency to pomalidomide (Pom). (**C**) In H9 hES cells with 20 μM Thal, Pom, and NAP treatments, SALL4 expression was measured and quantified relative to β-actin expression. *** *p* < 0.001 versus control value; ### *p* < 0.001 relative to the NAP value by Tukey’s multiple comparisons test. (**D**) In H9 cells, SALL4 protein expression was measured using Western blots, following 3 μM treatments of vehicle (control), Thal, Pom, 3,6′-DP, and 1,6′-DP. SALL4 expression was normalized to β-actin levels. 3,6′-DP and 1,6′-DP significantly elevated SALL4 levels relative to Pom treatment. * *p* < 0.05, **** *p* < 0.0001: treatments versus control (i.e., DMSO/vehicle); ## *p* < 0.01: treatments versus Pom of same concentration (values: mean ± SEM, *n* = 4/group). Panels A and C were adapted from Hsueh et al., 2021 [28]; Panels B and D were adapted from Tsai et al., 2022 [108].

**Figure 7 biomolecules-13-00747-f007:**
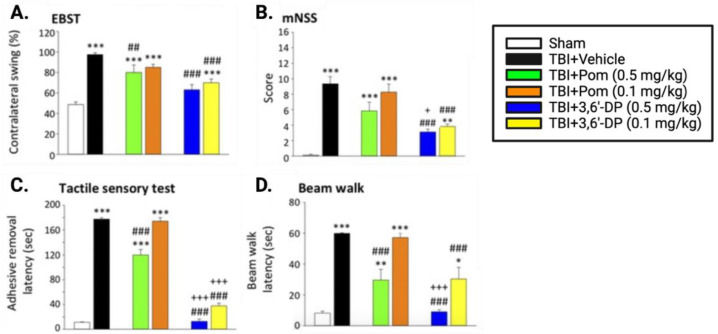
Behavioral assay data of Lin and colleagues (2020) reveal cognitive and behavioral benefits of 3,6′-DP. Scores of sham animals with no injury, TBI animals treated with vehicle (control), and TBI animals treated with 0.5 or 0.1 mg/kg body weight pomalidomide (Pom) or 3,6′-DP are shown 24 h post-TBI induction. We quantified body asymmetry using the EBST (**A**), neurological function using mNSS (**B**), sensorimotor function using a tactile sensory test (**C**), and motor coordination using a beam walk (**D**). Data represent the mean ± S.E.M. (*n* = 5 in each group). * *p* < 0.05, ** *p* < 0.01, *** *p* < 0.001 versus the Sham group; ## *p* < 0.01, ### *p* < 0.001 versus the TBI + Veh group; + *p* < 0.05, +++ *p* < 0.001 versus the TBI + Pom (0.5 mg/kg body weight) group. Image adapted from Lin et al., 2020 [105].

**Figure 8 biomolecules-13-00747-f008:**
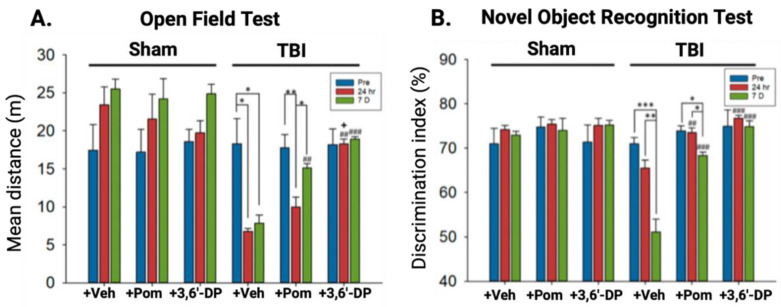
Behavioral assay data of Huang and colleagues (2021) demonstrate beneficial effects of 3,6′-DP on locomotion and anxiety (**A**) and short-term memory (**B**) 24 h and seven days post CCI TBI induction. (**A**) The mean distance traveled, expressed as the mean ± SEM, at 24-h and seven-day time points for sham and TBI animals treated with vehicle (Veh), pomalidomide (Pom), or 3,6′-DP. Both Pom and 3,6′-DP ameliorated TBI-induced locomotion deficits and anxiety-related behaviors at the seven-day time point; 3,6′-DP uniquely improved these behaviors at the 24-h time point. (**B**) The discrimination indices, expressed as the mean ± SEM, at 24-h and seven-day time points for sham and TBI animals treated with Veh, Pom, or 3,6′-DP. 3,6′-DP entirely remediated the TBI-induced short-term memory deficit seven days following injury; Pom only partially restored short-term memory function at the seven-day point. * *p* < 0.05, ** *p* < 0.01, *** *p* < 0.001 versus different time points in the same group; ## *p* < 0.01, ### *p* < 0.001 versus TBI + Veh group at the same time point. + *p* < 0.05 versus TBI + Pom group at the same time point (*n* = 5 in each group). Statistical differences were analyzed using one-way ANOVA, followed by Tukey’s post hoc comparisons. The image was adapted from Huang et al., 2021 [106].

**Figure 9 biomolecules-13-00747-f009:**
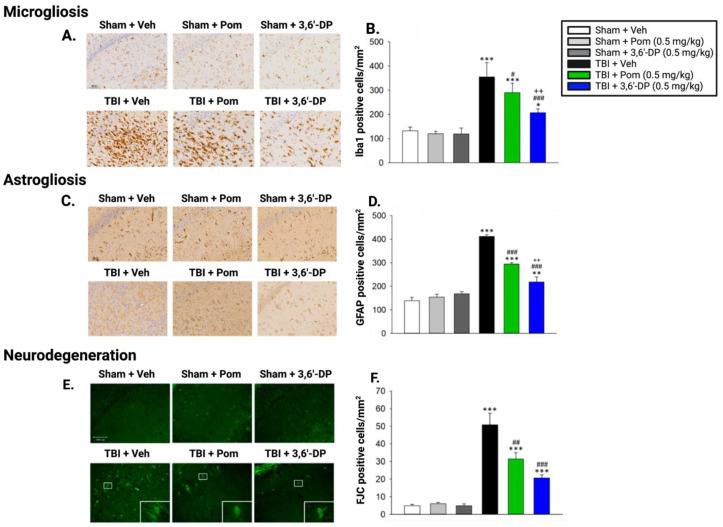
In the CA1 region of the hippocampus, 3,6′-DP and, to a lesser extent, pomalidomide (Pom) treatments mitigate TBI-induced neuroinflammation/microgliosis, as indicated by reduced levels of activated microglia (Iba1+), astrogliosis, as indicated by reduced levels of reactive astrocytes (GFAP+), and neurodegeneration, as indicated by reduced number of degenerating cells (FJC+). All photomicrographs and cell quantifications were taken seven days post TBI. All treatments were administered at a 0.5 mg/kg body weight dose. (**A**) Photomicrographs of Iba1-stained CA1 tissue (scale bar: 60 μm). (**B**) Iba1+ activated microglia quantified in the CA1 region. (**C**) Photomicrographs of GFAP-stained CA1 tissue (scale bar: 60 μm). (**D**) GFAP+ reactive astrocytes quantified in the CA1 region. (**E**) Photomicrographs of FJC-stained CA1 tissue (scale bar: 100 μm). (**F**) FJC+ degenerating cells quantified in the CA1 region. Data are expressed as mean ± S.E.M. * *p* < 0.05, ** *p* < 0.01, *** *p* < 0.001 versus Sham + Veh group; # *p* < 0.05, ## *p* < 0.01, ### *p* < 0.001 versus TBI + Veh group; ++ *p* < 0.01 versus TBI + Pom group (*n* = 5 per group). Statistical differences were analyzed using independent sample *t*-tests and one-way ANOVA, followed by Tukey’s post hoc comparisons. Image adapted from Huang et al., 2021 [106].

**Figure 10 biomolecules-13-00747-f010:**
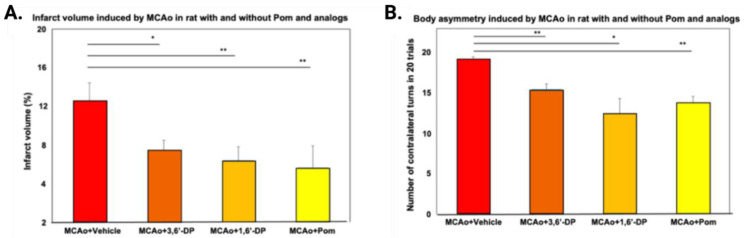
Improvements in stroke-induced infarct volume and body asymmetry with 3,6′-DP, 1,6′-DP and pomalidomide (Pom) treatments. Treatments (equimolar to 20 mg/kg body weight Pom) were administered via intraperitoneal (i.p.) injection following one hour of MCAo and 30 min of reperfusion. (**A**) Infarct volumes of MCAo rats treated with Pom and its dithionated analogs 3,6′-DP and 1,6′-DP. (**B**) Body asymmetry as quantified by number of contralateral turns in the EBST. Non-MCAo rats perform roughly equal numbers of ipsi- and contralateral turns; injured rats exhibit a bias toward contralateral turns. Data are expressed as mean + standard error of the mean (S.E.M., *n* = 4 per group), Dunnett’s *t*-test * *p* < 0.05, ** *p* < 0.01 versus MCAo + vehicle group. Image adapted from Tsai et al., 2022 [108].

**Figure 11 biomolecules-13-00747-f011:**
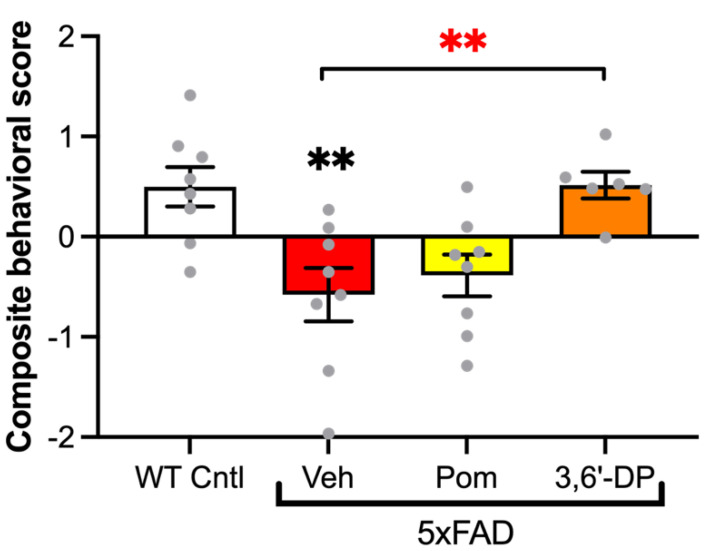
Cognitive behavioral markers ameliorated with 3,6′-DP treatment. Treatments were administered at 26.4 mg/kg body weight (pomalidomide (Pom)) or 29.5 mg/kg body weight (3,6′-DP) for four months. We calculated composite behavioral scores to increase the sensitivity between groups (WT control (Cntl), 5xFAD vehicle (Veh), 5xFAD Pom, and 5xFAD 3,6′-DP). ** *p* < 0.01 5xFAD veh versus WT cntl groups; ** *p* < 0.01 5xFAD veh versus 5xFAD 3,6′-DP groups by one-way ANOVA followed by Dunnett’s post-hoc tests. Image adapted from Lecca et al., 2022 [109].

**Figure 12 biomolecules-13-00747-f012:**
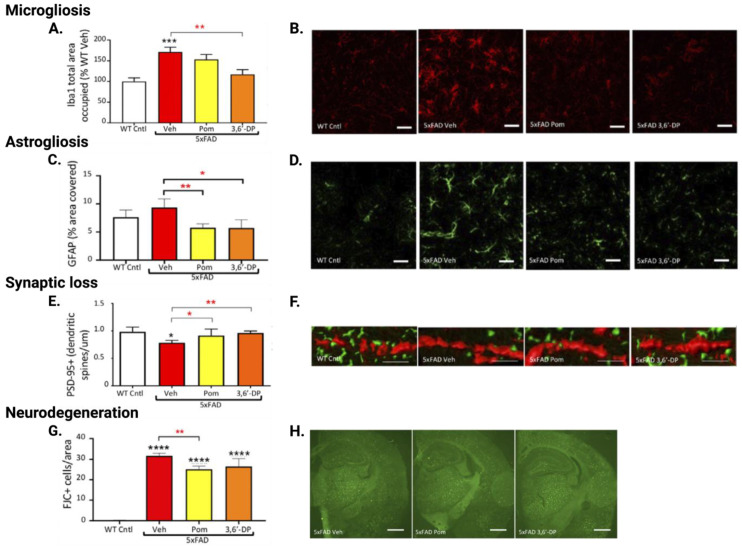
3,6′-DP and pomalidomide (Pom) mitigate microglial activation, reactive astrogliosis, synaptic loss, and neurodegeneration. 5xFAD mice were treated with vehicle (Veh), 29.5 mg/kg body weight 3,6′-DP, or 26.4 mg/kg body weight Pom i.p. daily for four months. (**A**) We quantified Iba1 total area occupied in immunohistochemical assays to measure activated microglia levels in the hippocampus. (**B**) Photomicrographs of Iba1+ cells (scale bar: 30 μm). (**C**) We used GFAP immunohistochemical assays to measure astrogliosis, expressing GFAP+ cells as a percentage of the brain area occupied. (**D**) Photomicrographs of GFAP+ cells (scale bar: 30 μm). (**E**) We quantified PSD-95+ dendritic spines as a measure of intact synapses in the hippocampus. (**F**) Representative images of PSD-95+ dendritic spines (green) located on MAP2+ dendrites (scale bar: 3 μm). (**G**) We quantified degenerating neurons by identifying FJC+ cells in the hippocampus. (**H**) Representative images of FJC-stained brain section (20×, tile scan over entire area of brain section, scale bar: 1 mm). * *p* < 0.05, *** *p* < 0.001, **** *p* < 0.0001 versus WT control (Cntl) group; * *p* < 0.05, ** *p* < 0.01 versus 5xFAD Veh group by one-way ANOVA followed by Dunnett’s post hoc test (*n* = 4–5 per group). Image adapted from Lecca et al., 2022 [109].

**Figure 13 biomolecules-13-00747-f013:**
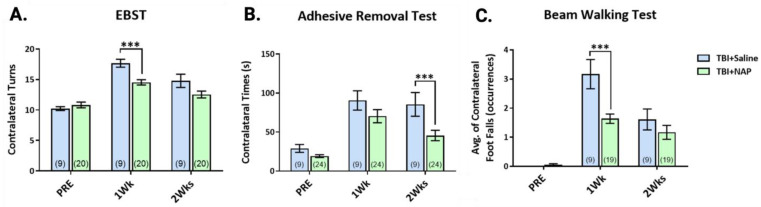
NAP treatment ameliorates TBI-induced behavioral impairments one or two weeks post-TBI. (**A**) The EBST was used to quantify body asymmetry. NAP treatment (30 mg/kg body weight i.p.) improved this measure one and two weeks post-TBI, although the difference at two weeks did not reach significance. (**B**) The ART was used to measure sensorimotor function, as TBI mice spend more time removing stickers from their contralateral paw than control mice. NAP treatment mitigated this impairment, reaching a significant difference at the two-week time point. (**C**) TBI mice demonstrated balance and motor coordination deficits in the BWT. NAP treatment reduced contralateral foot fall abnormalities associated with TBI, with the difference at one week reaching significance. Analysis by two-way ANOVA followed by Bonferroni’s test: *** *p* < 0.001. Data are expressed as mean ± SEM; *n* = 9 (TBI + Saline), *n* = 24 (TBI + NAP). Image adapted from Hsueh et al., 2021 [28].

**Figure 14 biomolecules-13-00747-f014:**
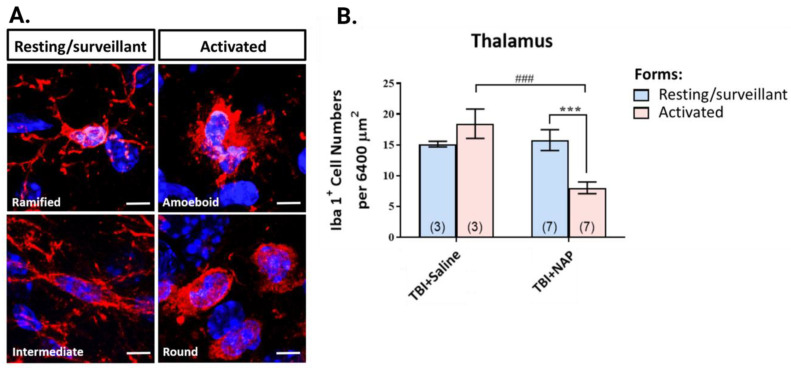
NAP treatment two weeks post TBI reduced microglial activation and promoted resting morphologies in the ipsilateral (injured) thalamus of TBI mice. (**A**) We utilized Iba1 immunofluorescence staining to image microglia in four forms: ramified, intermediate, amoeboid, and round. We classified each morphological form using the following criteria: (1) activated: amoeboid and round forms; (2) resting/quiescent: ramified long branching processes with a small cell body; and (3) intermediate transition forms. Scale bar: 5 μm. (**B**) Numbers of microglia in activated versus resting states in the ipsilateral thalamus. NAP treatment (30 mg/kg body weight i.p.) substantially decreased the number of microglia in active form relative to the TBI + saline group. *** *p* < 0.001, resting versus active form within TBI + NAP group; ### *p* < 0.001, active form of microglia in TBI + saline versus TBI + NAP groups. Two-way ANOVA with Bonferroni’s *t*-test for multiple comparisons. Image adapted from Hsueh et al., 2021 [28].

**Table 2 biomolecules-13-00747-t002:** Cytokine levels (pg/mL) in sham and MCAo rat plasma following pomalidomide (POM) and POM dithionated analog 3,6′-DP and 1,6′-DP treatments. POM, 3,6′-DP and 1,6′-DP demonstrate ability to improve cytokine-related neuroinflammation produced by stroke. Treatments (20 mg/kg body weight POM, 21.25 mg/kg body weight 3,6′-DP or 1,6′-DP) were administered via i.p. injection following one hour of MCAo and reperfusion. Plasma was collected 24 h post MCAo, and cytokine levels were quantified utilizing the Milliplex platform. All outcomes were compared to the MCAo + vehicle group (Dunnett’s *t*-test: * *p* < 0.05, ** *p* < 0.01 versus MCAo + vehicle group). TNF-α in the MCAo + 1,6′-DP group fell below the assay minimum detection level (2.22 pg/mL). The standard ranges for the assays were IL-1β: 2.4–10,000 pg/mL, TNF-α: 2.4–10,000 pg/mL, and IL-10: 7.3–30,000 pg/mL. Values are expressed as mean + S.E.M. (*n* = 4/group). Vehicle is DMSO alone. Adapted from Tsai et al., 2022 [108].

(N = 4/group)	Sham Control	MCAo + Vehicle	MCAo + 3,6′-DP	MCAo + 1,6′-DP	MCAo + POM
**IL-1β** (pg/mL)	6.58 ± 2.08	10.20 ± 0.76	4.06 ± 0.95 **	2.93 ± 0.21 **	4.76 ± 1.01 *
**TNF-α** (pg/mL)	4.08 ± 0.88 *	7.62 ± 1.04	3.39 ± 0.15 **	<2.22	2.59 ± 0.14 **
**IL-10** (pg/mL)	14.28 ± 3.16	12.51 ± 0.76	26.49 ± 0 86 **	11.59 ± 3.78	16.62 ± 3.21

## Data Availability

Not applicable.

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
