# Peer review of "A New Generation of IMiDs as Treatments for Neuroinflammatory and Neurodegenerative Disorders"

_biomolecules, 2023, doi:10.3390/biom13050747_

Round 1

Reviewer 1 Report

Very interesting and comprehensive review of thalidomide analogues as potential agents for reducing neurodegeneration. It is very well written and logical. The figures are good and the data is well presented. 

Author Response

We thank Reviewer 1 for their generous and supportive comments. No changes to our manuscript were requested.

Reviewer 2 Report

The paper describes the application of IMiDs in mediating neuroinflammation through cereblon-dependent and independent mechanisms. In general, immunomodulatory drugs are orally deliverable, have goo bioavailability, and making them more accessible and effective treatments for cancer treatments. Selective IMiDs can also penetrate blood brain barrier. However, the authors highlight the teratogenic effects associated with thalidomide and its analogues and highlight research work where better IMiDs with anti-inflammatory properties are developed. According to the authors, these new-generation IMiDs bind less with cereblon and fail to degrade SALL4, however, they retain or even enhance the anti-inflammatory property of the drug. These new IMiDs demonstrated slowing of the neurodegenerative disease progression and improved cognitive and behavioral outcomes.  

Interestingly, based on early work done by other groups, the authors cite ("AL Moreira et al., 1993") that IMiDs also reduce the expression of the pro-inflammatory cytokine TNF-α via cereblon-independent mechanism by binding to the 3’ untranslated region (UTR) of TNF-α mRNA and destabilize their mRNA, ultimately resulting in reduced translation into protein. The authors designed new generation IMiD analogues with Phthalimide lacking the glutarimide group of thalidomide that is essential to cereblon binding and showed TNF-α-lowering actions. TNF-α is a promising pharmacological target of neuroinflammatory drugs and a potential oral-based treatment for neurodegeneration will be highly preferred. And IMiDs (particularly pomalidomide) can be repurposed and used for CNS indications. Great thought.

As a protein degradation researcher who is working on molecular glues, this review is fascinating and thought-provoking. I enjoyed reading the work.  

I have some minor suggestions and curious questions.

  1. Is the new proposed molecule follows the same mechanism of action to produce the anti-inflammatory effect (degradation of TNF in the mRNA level)? Could you discuss if there are any qPCR experiments done with 3,6' DP and DNP? What would the authors suggest to further improve the molecule by design?
  2. Your figures and tables are good. However, can you add one extra column in table 1 and highlight if the proposed new IMiDs could cross the BBB? (along with CRBN binding or not, degradation or not, add BBB (can cross or not, that would be helpful).
  3. You have added an adamantyl group in your DNP IMiD scaffold, will this group affect its bioavailability and oral delivery?
  4. Just curious, what is the clinical status of these proposed drugs? Is preclinical studies all done? Please highlight what could be some of the adverse events you would expect with these new type of IMiDs.
  5. Since these IMiDs work more in a cereblon-independent fashion could you discuss what are the resistance mechanism one could expect from this molecule?
  6. More of a comment, you have compared your proposed IMiDs with pomalidomide for all your TBI experiments. I personally think a better control for these experiments would be antibodies that target TNF-a (infliximab, etanercept or other TNF inhibitors) vs DNP and/or 3,6'DP. Please comment.
  7. If there is a possibility, please reduce the size of the overall manuscript. I find the paper is a little longer than necessary.

In summary, developing a new generation IMiD with the mechanism of avoiding cereblon binding and neosubstrate degradation would potentially prevent the risk of teratogenicity, the authors have chosen to investigate the cereblon pathway-independent anti-inflammatory properties of IMiDs in the treatment of neuroinflammation and CNS diseases. I find the thought is creative at least in the protein degradation field. The article is written in a comprehensible way. I highly recommend this work for publication. 

Congratulations to the authors.

Author Response

We thank Reviewer 2 for their thoughtful comments and generous support. We have responded to their concerns/suggestions in the attached document.

Reviewer 3 Report

This is a well written manuscript that overviews the immunomodulatory imide drug (IMiD) class and their potential to change the treatment of inflammatory-associated conditions, including neurological disorders. The manuscript also highlights the potential of novel non-classical IMiDs designed to avoid binding with human cereblon and/or evade degradation of downstream neosubstrates, thus providing new hope for the development of effective treatments for erythema nodosum leprosum (ENL) and neurodegenerative disorders. Overall, this manuscript provides a positive outlook on the potential of the IMiD class of drugs to improve patient outcomes and represents an exciting area of research with enormous therapeutic potential. While the manuscript mentions the teratogenic effects of classical IMiDs, it would be informative to include more information on the specific risks and side effects associated with the new non-classical IMiDs, including any potential long-term effects and the challenges of developing and bringing new drugs to market.

Author Response

We thank Reviewer 3 for their kind comments and thoughtful question re: potential teratogenic/adverse actions of new classical and atypical IMiDs - which is responded to under Q8 in the attachment - we modified our manuscript (Conclusions - final page of text) to include new text relating to such adverse/teratogenic events.
